# On the Effectiveness of Persistent Homology

**Renata Turkeš**
University of Antwerp
`renata.turkes@uantwerpen.be`

**Guido Montúfar**
University of California, Los Angeles
`montufar@math.ucla.edu`

**Nina Otter**
Queen Mary University of London
`n.otter@qmul.ac.uk`

## Abstract

Persistent homology (PH) is one of the most popular methods in Topological Data Analysis. Even though PH has been used in many different types of applications, the reasons behind its success remain elusive; in particular, it is not known for which classes of problems it is most effective, or to what extent it can detect geometric or topological features. The goal of this work is to identify some types of problems where PH performs well or even better than other methods in data analysis. We consider three fundamental shape analysis tasks: the detection of the number of holes, curvature and convexity from 2D and 3D point clouds sampled from shapes. Experiments demonstrate that PH is successful in these tasks, outperforming several baselines, including PointNet, an architecture inspired precisely by the properties of point clouds. In addition, we observe that PH remains effective for limited computational resources and limited training data, as well as out-of-distribution test data, including various data transformations and noise. For convexity detection, we provide a theoretical guarantee that PH is effective for this task in $\mathbb{R}^d$, and demonstrate the detection of a convexity measure on the FLAVIA data set of plant leaf images. Due to the crucial role of shape classification in understanding mathematical and physical structures and objects, and in many applications, the findings of this work will provide some knowledge about the types of problems that are appropriate for PH, so that it can — to borrow the words from Wigner 1960 — "remain valid in future research, and extend, to our pleasure", but to our lesser bafflement, to a variety of applications.

## 1 Introduction

Persistent homology (PH) is an extension of homology, which gives a way to capture topological information about connectivity and holes in a geometric object. PH can be regarded as a framework to compute representations of raw data that can be used for further processing or as inputs to learning algorithms. There have been numerous successful applications of PH in the last decade, from prediction of biomolecular properties [18, 19, 110], face, gait and activity recognition [58, 66, 70, 122] or digital forensics [6], to discriminating breast-cancer subtypes [97], quantifying the porosity of nanoporous materials [69], classifying fingerprints [50], or studying the morphology of leaves [71].[1] At the same time, the reasons behind these successes are not yet well understood. Indeed, the data used in real-world applications is complex, so that there are numerous effects at play and one is often left unsure why PH worked, i.e., what type of topological or geometric information it captured that facilitated the good performance.

---

[1] A database of applications of persistent homology is being maintained at [51].

36th Conference on Neural Information Processing Systems (NeurIPS 2022).

The title of our manuscript is inspired by a famous paper from 1960, "The unreasonable effectiveness of mathematics in the natural sciences" [112], in which Wigner discusses, with wonder, how mathematical concepts have applicability far beyond the context in which they were originally developed. The same, we believe, is true for persistent homology. While this method has been applied successfully to a wide range of application problems, we believe that for PH to remain relevant, there is a need to better understand why it is so successful. Thus, we distinguish between the *usefulness* of PH for applications, which has been attested in hundreds of applications and publications, and its *effectiveness*, namely that PH is capable of producing an intended or desired result. Thus, here we initiate an investigation into the effectiveness of PH, or in other words, we investigate *what* is seen by persistent homology: Given a data set, i.e., a point cloud, which underlying topological and geometric features can we detect with PH? This question is related to manifold learning and, specifically, topological and geometric inference: Given a finite point cloud $X$ of (noisy) samples from an unknown manifold $M$, how can one infer properties of $M$ [11, 12, 25, 29]? Obtaining a representation of a shape that can be used in statistical models is an important task in data analysis and numerous approaches to modeling surfaces and shapes [104].

To pursue our investigation, we set out to identify some fundamental data-analysis tasks that can be solved with PH. Since PH is inspired by homology, which provides a measure for the number of components, holes, voids, and higher-dimensional cycles of a space — to which we collectively refer as "topological features" —, we start with the obvious question of whether PH applied to a point cloud sampled from a geometric object can detect the **number of (1-dimensional) holes** of the underlying object. Unlike homology, however, PH registers also the persistence of topological features across scales, and can thereby capture geometric information, such as size or position of holes. We therefore also investigate how well PH can detect fundamental geometric notions of **curvature** and **convexity**. For each of the three problems, we first discuss theoretical results that provide a guarantee that PH can solve these tasks. Detection of convexity with PH has not been investigated in the literature to date, and we prove a new result. To investigate how well the PH pipeline works in practice, we compare its performance against several baselines on synthetic point-cloud data sets. As a first machine learning (ML) baseline we take an SVM trained on the distance matrices of point clouds. We further consider fully-connected neural networks (NN) with a single or multiple hidden layers, also trained on distance matrices. As a stronger baseline we consider a PointNet trained on the point clouds directly. PointNet [1, 88] is designed specifically for point cloud data. Similar architectures with convolutional (and fully-connected and pooling) layers have been applied for Betti-number and curvature estimation [52, 81]. For convexity detection, we also evaluate the performance of PH on real-world data. The theoretical guarantees above imply that the results for PH would generalize to new data.

Finally, we note that our goal is not to claim the superiority of PH compared to other approaches in the literature, in particular, with the state-of-the-art methods for each of the problems. We do not necessarily expect that on well-specified mathematical problems PH will beat state-of-the-art algorithms that have been specifically designed for those tasks. Instead, what we think is interesting and remarkable is that PH can in fact solve tasks it is not specifically or uniquely designed for. Moreover, an advantage of PH is that it can reveal, e.g., both topology and curvature at the same time, avoiding the need to employ and combine state-of-the-art models for each of the tasks.

**Related work** In spite of the growing interest in PH, so far there is only limited work in the direction that we pursue here. There is indeed theoretical evidence that the number of holes of the underlying space can be detected from PH (under some conditions about the target space, the sample density and closeness to the space) [32, 63, 77], and there is significant interest in investigating how well this works in practice [30]. However, so far there are only few available results. Some works demonstrate that PH can be used to detect the number of holes, but only on individual toy examples (e.g. [90], [25, Figure 19], [65, Figures 9-20], [29, Figures 2, 3, 6, 11, 12]) without looking into the statistical significance between different classes of data, or the accuracy of some classification algorithms on a comprehensive data set. There are also some works where PH is used to estimate the Betti numbers on a possibly larger data set, but only with the goal of using this information to e.g., study the behavior of deep neural networks [76] or ensure topologically correct dimensionality reduction [82] or image segmentation [56], so that the soundness of this estimation is not investigated, which is the focus of our work. Some insights about PH and curvature have been obtained in the literature, starting with an illustrative example in [39, Figure 12] which shows that PH on the filtered tangent complex can distinguish between letters (C and I) that have the same topology, since their

curvature is different. Recently, [17] show both theoretically and experimentally that PH can predict curvature (with computational experiments replicated in [107]), which inspired us to investigate this problem in more detail. Regarding the important geometric problem of classification between convex and concave shapes, we were not able to identify any previous works investigating the applicability of PH to this task.

Some further recent work investigating the topological and geometric features seen by PH are the following. Bubenik and Dłotko [16] show that using PH of points sampled from spheres one can determine the dimension of the underlying spheres. A connection has also been established between PH and the magnitude of a metric space (an isometric invariant) [79]. There have been several efforts in using PH to estimate fractal dimensions, such as [95] in which Schweinhart proves that the fractal dimension of some metric spaces can be recovered from the PH of random samples.

**Main contributions**   Our contributions can be summarized as follows.

- We prove that PH can detect convexity in $\mathbb{R}^d$ (Theorem 1).
- We define a new tubular filtration function (medium through which PH is extracted from data), that is crucial for the detection of convexity (Definition 1).
- We demonstrate experimentally that PH can detect the number of holes (Section 3), curvature (Section 4), and convexity (Section 5) from synthetic point clouds in $\mathbb{R}^2$ or $\mathbb{R}^3$, outperforming SVMs and fully-connected networks trained on distance matrices, and PointNet trained on point clouds. For convexity detection, we also show that PH obtains a good performance on a real-world data set of plant leaf images.
- We demonstrate experimentally that PH features allow to solve the above tasks even in the case of limited training data (Section 3), noisy (Section 3) and out-of-distribution (Section 5) test data, and limited computational resources (Section 3, Section 4, Section 5).
- We provide insights about the topological and geometric features that are captured with long and short persistence intervals (Section 6), and formulate guidelines for applications that are suitable for PH (Section 7).
- We provide data sets that can be directly used as a benchmark for our tasks or other related point-cloud-analysis or classification problems. We provide computer code to construct more data and replicate our experiments.

## 2   Background on persistent homology

Homology is a topological concept that attempts to distinguish between topological spaces by constructing algebraic invariants that reflect their connectivity properties [90], i.e., $k$-dimensional cycles (components, holes, voids, ...). The number of independent $k$-dimensional cycles is called $k$-th Betti number and denoted by $\beta_k$. For example, the circle has Betti numbers $\beta_0 = 1$, $\beta_1 = 1$, $\beta_2 = 0$, and for a torus, we have $\beta_0 = 1$, $\beta_1 = 2$, $\beta_2 = 1$.

Persistent homology is an extension of this idea [123] that has found success in applications to data. To calculate PH from some data $X$, we must first build a filtration, i.e., a family of nested topological spaces $\{K_r\}_{r \in \mathbb{R}}$ which, in a suitable sense, approximate $X$ at different scales $r \in \mathbb{R}$. Typically, $X = \{x_1, x_2, \dots, x_n\}$ is a point cloud in $\mathbb{R}^d$, and $K_r$ is a simplicial complex, a set of simplices $\sigma$ (which we can think of as vertices, edges, triangles, ...) such that if $\sigma \in K_r$ and $\tau \subseteq \sigma$, then $\tau \in K_r$ [39]. A common choice is $K_r = VR(X, r)$, where $VR(X, r)$ is the Vietoris-Rips simplicial complex, in which $\sigma = \{x_1, \dots, x_m\} \in VR(X, r)$ when $\mathrm{dist}(x_i, x_j) \leq r$ for all $1 \leq i, j \leq m$. PH can then be summarized with a persistence diagram (PD), a scatter plot with the $x$ and $y$ axes respectively depicting the scale $r \in \mathbb{R}$ at which each cycle is born and dies or is identified (i.e., merges) with another cycle within a filtration. The length $l = d - b$ of a persistence interval $(b, d)$ measures the lifespan — the so-called "persistence" — of the corresponding cycle in the filtration.

Instead of working directly with persistence diagrams, these are often represented by other signatures that are better suited for machine learning frameworks. A common choice is a persistence image (PI) [2] (discretized sum of Gaussian kernels centered at the PD points), or a persistence landscape (PL) (functions obtained by "stacking isosceles triangles" above persistence intervals, with height reflecting their lifespan) [15]. The steps for extracting PH features are visualized in Appendix B.2. For good choices of filtration and signature [103], there are theoretical results that guarantee that PH

is stable under small perturbations [26, 98]. After the PH signature is calculated, statistical hypothesis testing [10, 15], or machine learning techniques such as SVM or k-NN [2, 48, 78] can be used on these features to study the differences within the data set of interest. It is important to note that PH is very flexible, as different choices can be made in every step of the pipeline, regarding the input and output of PH, as detailed in the remainder of this section.

## 2.1 Approximation of a space at scale $r \in \mathbb{R}$

Instead of the Vietoris-Rips complex, other types of complexes can be used to approximate the data $X$ at the given scale $r \in \mathbb{R}$. For instance, since Vietoris-Rips simplicial complex is large [80], one might rather choose the alpha complex [46] (for a visualization, see Appendix C.2), which is closely related to the Vietoris-Rips complex [63], but consists of significantly less simplices and is faster to construct when the dimension of the ambient space is 2 or 3 (for details, see Appendix A.4). If data $X$ is an image rather than a point cloud, or if the point cloud can be seen as an image without losing important information, cubical complexes [59] might be a more suitable choice, where vertices, edges and triangles are replaced by vertices, edges and squares (for a visualization, see Appendix E.1).

## 2.2 Filtration

A filtration $\{K_r\}_{r \in \mathbb{R}}$ of a point cloud in $\mathbb{R}^d$ can be constructed from any function $f : \mathbb{R}^d \to \mathbb{R}$ by considering $K_r$ to be the sublevel set of $f$ thresholded by $r \in \mathbb{R}$: $\{y \in \mathbb{R}^d \mid f(y) \le r\}$. The underlying filtration function in the common PH pipeline introduced above is the distance function $\delta_X : \mathbb{R}^d \to \mathbb{R}$, where $\delta_X(y) = \min\{\text{dist}(y, x) \mid x \in X\}$ is the distance to point cloud $X$. Indeed, the Vietoris-Rips simplicial complex $VR(X, r)$ approximates the sublevel set

$$K_r = \delta_X^{-1}((-\infty, r]) = \{y \in \mathbb{R}^d \mid \delta_X(y) \le r\} = \cup_{x \in X} B(x, r),$$

where $B(x, r)$ is a ball with radius $r$ centered around $x \in X$ [27].

However, PH on such a filtration is very sensitive to outliers, since even a single outlier changes $\delta_X$ significantly. In the presence of outliers, it is better to replace the distance function with the Distance-to-Measure (DTM) $\delta_{X,m} : \mathbb{R}^d \to \mathbb{R}$, where $\delta_{X,m}(x)$ is the average distance from a number of neighbors on the point cloud [5, 27] (for a visualization, see Appendix C.2). However, depending on the task, there are many other filtration functions one could choose, such as rank [84], height, radial, erosion, dilation [48], and the resulting PH captures completely different information about the cycles [103]. For example, whereas PH with respect to the Vietoris-Rips filtration encodes the size of the hole, PH on the height filtration informs about the position of the hole. For a more detailed discussion about the influence of filtration, see Appendix F.

## 2.3 Persistence signature

Next to PIs and PLs, a plethora of persistence signatures has been introduced in the literature, e.g., Betti numbers [57, 106] or Euler characteristic [72] (across scales), or even scalar summaries such as amplitude [48], entropy [35, 93], or algebraic functions of the birth and death values [4, 62]. Some of these signatures summarize the same information, but lie in different metric spaces [105]. Others, however, such as the scalar summaries listed above, discard information compared to PDs, as it might sometimes be useful to, e.g., only capture the (total or maximum) persistence of intervals, but not all the detailed information about all birth and death values (see Appendix F).

# 3 Number of holes

In this section, we focus on the task of (ordinal) classification of point clouds by the number of 1-dimensional holes. Research in psychology shows that global properties often dominate perception, and, in particular, that topological invariants such as number of holes, inside versus outside, and connectivity can be effective primitives for recognizing shapes [85]. Extracting such topological information can therefore prove useful for many computer vision tasks. There are theoretical results in the literature that ensure that PH with respect to the alpha simplicial complex can be successful for this problem (Appendix A.2), and the computational experiments that follow demonstrate this success in practice.

**Data**  We consider 20 different shapes in $\mathbb{R}^2$ and $\mathbb{R}^3$, with four different shapes having the same number of holes $(0, 1, 2, 4$ or $9)$. For each shape, we construct 50 point clouds each consisting of $1\,000$ points sampled from a uniform distribution over the shape, resulting in a balanced data set of $1\,000 = 20 \times 50$ point clouds. A few examples of these point clouds are shown in Figure 1. The label of a point cloud is the number of holes in the underlying shape.

| shapes | | | | | |
|---|---|---|---|---|---|
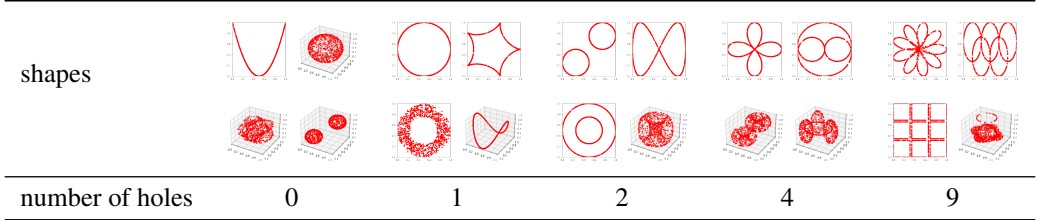
| number of holes | 0 | 1 | 2 | 4 | 9 |

Figure 1: Number of holes data set.

**PH pipeline**  For each point cloud $X$ and scale $r \in \mathbb{R}$, we consider its alpha complex. We will look into scenarios in which data contains noise, and therefore, instead of the standard distance function, we consider Distance-to-Measure (DTM) as the filtration function. We extract 1-dimensional PDs, that are then transformed to PIs, PLs, or a simple signature consisting only of lifespans $l = d - b$ of the 10 most persisting cycles (as there are at maximum 9 holes of interest in the given data set)[2], and classified with an SVM. We consider a "PH simple" pipeline, which relies on the 10 lifespans, and a "PH" pipeline wherein grid search is employed to choose the best out of the three aforementioned persistent signatures and the values of their parameters. For more details on the pipeline, see Appendix B.2 and Appendix C.2.

**Results**  We investigate the clean and robust test accuracy under four types of transformations (translation, rotation, stretch, shear) and two types of noise (Gaussian noise, outliers). For more details on these transformations, see Appendix C.1. We train the classifier on $80\%$ of the original point clouds, and test on the remaining $20\%$ of the data either in its original form or subject to transformations and noise. The results reported in Figure 2 (with detailed results across multiple runs in Appendix C.3) show that PH obtains very good test accuracy on this classification task, even in the presence of affine transformations or noise, outperforming baseline machine- and deep-learning techniques.[3]  We reach a similar conclusion in case of limited training data and computational resources (Appendix C.4, Appendix C.6). Firstly, the evolution of the test accuracy across different amounts of training data demonstrates that PH achieves good performance for a small number of training point clouds, which is not the case for other pipelines. Secondly, although the hyperparameter tuning of the PH pipeline does take time (as we consider a wide range of parameters for the different persistence signatures), it is still less than for PointNet. Moreover, Figure 2 shows that even the simple PH pipeline, where the SVM is used directly on the lifespans of the 10 most persisting cycles (without any tuning of PH-related parameters) performs well.

## 4   Curvature

This section considers a regression task to predict the curvature of an underlying shape based on a point cloud sample. Estimating curvature-related quantities is of prime importance in computer vision, computer graphics, computer-aided design or computational geometry, e.g., for surface segmentation, surface smoothing or denoising, surface reconstruction, and shape design [23]. For continuous

---

[2]Although a sphere has no 1-dimensional holes, its PD might consist of many short intervals which correspond to the small holes on the surface. In addition, in the presence of noise, additional small holes might appear for any point cloud. Hence, it is not a good idea to consider the cardinality |PD| of the PD as the signature.

[3]Interestingly, although PointNet was designed with the idea to be invariant to affine transformations, it performs poorly when the test data is translated or rotated (and this is consistent with some previous results [68, 111, 115, 117–120]), or when it contains outliers. Traditional neural networks perform very poorly, which might not come as a big surprise, since it was recently demonstrated that they transform topologically complicated data into topologically simple one as it passes through the layers, vastly reducing the Betti numbers (nearly always even reducing to their lowest possible values: $\beta_k = 0$ for $k > 0$, and $\beta_0 = 1$) [76]. Of course, the choice of activation function and hyperparameters might have an important influence on performance [76].

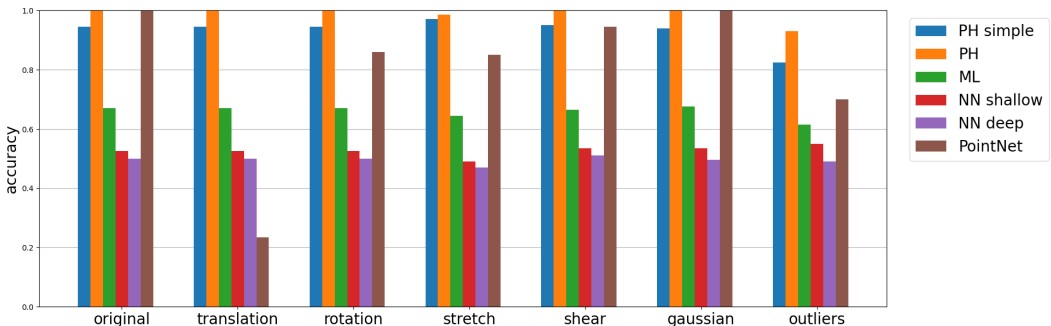

Figure 2: Persistent homology can detect the number of holes.

surfaces, normals and curvature are fundamental geometric notions which uniquely characterize local geometry up to rigid transformations [52]. Recently, it has been shown that, using PH, curvature can be both recovered in theory (Appendix A.3), and effectively estimated in practice [17]. We run a similar experiment, evaluating the PH pipeline against our baselines, and also taking a closer look into the importance of short intervals.

**Data** A balanced data set is generated in the same way as in [17]: We consider unit disks $D_\kappa$ on surfaces of constant curvature $\kappa$: (i) $\kappa = 0$, Euclidean plane, (ii) $\kappa > 0$, sphere with radius $1/\sqrt{\kappa}$, and (iii) $\kappa < 0$, Poincaré disk model of the hyperbolic plane. Curvature $\kappa$ lies in the interval $[-2, 2]$ so that a disk with radius one can be embedded on the upper hemisphere of a sphere with constant curvature $\kappa$ (as it spherical cap). For each $\kappa \in \{-2, -1.96, \ldots, -0.04, 0, 0.04, \ldots, 1.96\}$, we construct 10 point clouds by sampling 500 points from the unit disk $D_\kappa$ with the probability measure proportional to the surface area measure [15, Section 2.7, Section 4.1]. A few examples with $\kappa \in \{-2, -1, -0.1, 0, 0.1, 1, 2\}$ are illustrated in Figure 3[4].These $101 \times 10 = 1\,010$ point clouds are considered as the training data, whereas the test data set is built in a similar way for 100 values of $\kappa$ chosen uniformly at random from $[-2, 2]$. The label of a point cloud is the curvature $\kappa$ of the underlying disk $D_\kappa$. Note that all these disks are homoemorphic: they are contractible, so that their homology is trivial and homology is thus unable to distinguish between them [17].

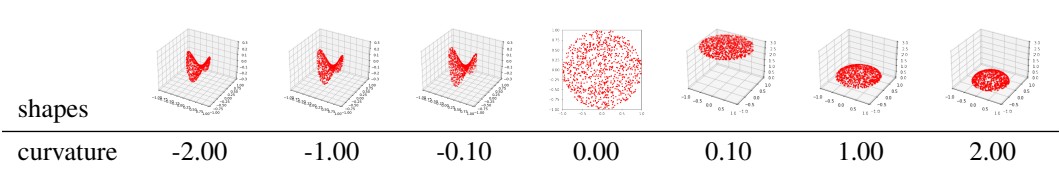

| shapes | | | | | | | |
|--------|---|---|---|---|---|---|---|
| curvature | -2.00 | -1.00 | -0.10 | 0.00 | 0.10 | 1.00 | 2.00 |

Figure 3: Curvature data set.

**PH pipeline** For each point cloud $X$, we first calculate the suitable matrix of pairwise distances between the point-cloud points: hyperbolic, Euclidean or spherical, respectively for negative, zero and positive curvature [15, Section 2.7]. The input for PH is the filtered Vietoris-Rips simplicial complex.[5] We extract 0- and 1-dimensional PDs, which are then transformed into PIs, PLs or lifespans, to be fed to SVM. More details on the pipeline are provided in Appendix B.2 and Appendix D.1.

**Results** Figure 4 shows the mean squared errors for the PH and other pipelines, together with their regression lines, with detailed results across multiple runs listed in Appendix D.2. The results show that PH indeed detects curvature, outperforming other methods.[6] Next to the PH pipeline discussed

---

[4]The unit disks with negative curvature are here visualized on hyperbolic paraboloids. These saddle surfaces have non-constant curvature, but they locally resemble the hyperbolic plane.

[5]Alpha complex is faster to compute, but involves Delaunay triangulation, whose unique existence is guaranteed only in Euclidean spaces. To calculate PH, we rely on the Ripser software [102], which is at the time the most efficient library to compute PH with Vietoris-Rips complex [80].

[6]Simple machine and deep learning techniques are able to differentiate between positive and negative curvature, but perform poorly in predicting the actual value of the curvature of the underlying surface.

above, wherein a grid search is used to tune the parameters (Appendix B.2), we also consider SVM on the lists of lifespans of all persistence intervals (PH simple), and SVM only on the 10 longest lifespans (PH simple 10), in order to investigate if all persistence intervals contribute to prediction. We see that the performance drops if we only focus on the longest 10 intervals, so that the many short intervals together capture the geometry of interest for this problem. Similarly as in Section 3, the grid search across the different parameters for persistence signatures does take time (Appendix D.3), but Figure 4 shows that SVM on a simple signature of all (0-dimensional) lifespans performs well. We highlight that the data used here, as was the data in Bubenik's work [17], is sampled from surfaces with *constant* curvature. In future work it would be interesting to conduct similar experiments on shapes with non-constant curvature.

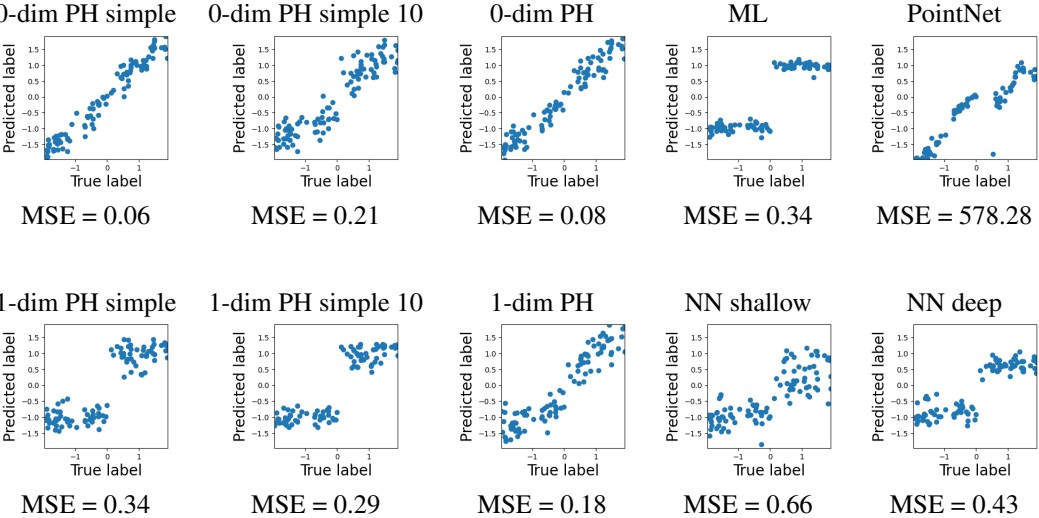

Figure 4: Persistent homology can detect curvature.

## 5   Convexity

In this section, we consider the binary classification task that consists of detecting whether a point cloud is sampled from a convex set. Convexity is a fundamental concept in geometry [40], which plays an important role in learning, optimization [9], numerical analysis, statistics, information theory, and economics [91]. Furthermore, points of convexities and concavities have been demonstrated as crucial for human perception of shapes across many experiments [94].

To the best of our knowledge, prior to our work PH has not been employed to analyze convexity, and it is a task for which PH's effectiveness might seem surprising. In the first decade after the introduction of PH, it was seen primarily as the descriptor of global topology. Recently, there have been many discussions and greater understanding that PH also captures local geometry [3]. However, it is still suggested that the long persistence intervals capture topology (as was the case with the detection of holes in Section 3), and many—even too many for the human eye to count—short persistence intervals capture geometrical properties (as was the case with curvature prediction in Section 4). However, as we show in Theorem 1 (proof in Appendix A.1) and as our experiments suggest, it is a single, and the second-longest persistence interval that enables us to detect concavity. A crucial ingredient in our result is the introduction of tubular filtrations (Definition 1), which, to the best of our knowledge, are a novel contribution to the TDA literature (details in Appendix A.1).

**Definition 1.** Given a line $\alpha \subset \mathbb{R}^d$, we define the **tubular function with respect to** $\alpha$ as follows:

$$\tau_\alpha \colon \mathbb{R}^d \to \mathbb{R}$$
$$x \;\mapsto \mathrm{dist}(x, \alpha),$$

where $\mathrm{dist}(x, \alpha)$ is the distance of the point $x$ from the line $\alpha$. Given $X \subset \mathbb{R}^d$ and a line $\alpha$, we are interested in studying the sublevel sets of $\tau_\alpha$, i.e., the subsets of $X$ consisting of points at a specific distance from the line. We define

$$X_{\tau_\alpha, r} = \{x \in X \mid \tau_\alpha(x) \le r\} \;= \{x \in X \mid \mathrm{dist}(x, \alpha) \le r\}\,.$$

We call $\{X_{\tau_\alpha, r}\}_{r \in \mathbb{R}_{\geq 0}}$ the **tubular filtration with respect to** $\alpha$.

**Theorem 1.** Let $X \subset \mathbb{R}^d$ be triangulizable. We have that $X$ is convex if and only if for every line $\alpha$ in $\mathbb{R}^d$ the persistence diagram in degree 0 with respect to the tubular filtration $\{X_{\tau_\alpha, r}\}_{r \in \mathbb{R}_{\geq 0}}$ contains exactly one interval.

**Data** We construct a balanced data set by sampling $5\,000$ points from convex and concave (non-convex) shapes in $\mathbb{R}^2$. First, we consider the "regular" convex shapes of triangle, square, pentagon and circle, and their concave variants, sampling 60 point clouds of each of the eight shapes, 480 point clouds in total. Next, we build 480 "random" convex and concave shapes, in order to be able to investigate if an algorithm is actually detecting convexity, or only the different basic shapes. A few examples are shown in Figure 5. To construct a random convex shape, we generate 10 points at random, and then build their convex hull using the quickhull algorithm [108]. We construct random concave shapes in a similar way, but instead of the convex hull, we build the alpha shape [45, 47] with the optimized alpha parameter, which gives a finer approximation of a shape from a given set of points. If the alpha shape is convex (i.e., if the alpha shape and its convex hull are the same), we reconstruct the concave shape from scratch. A point cloud has label 1 if it is sampled from a convex shape, and 0 otherwise.

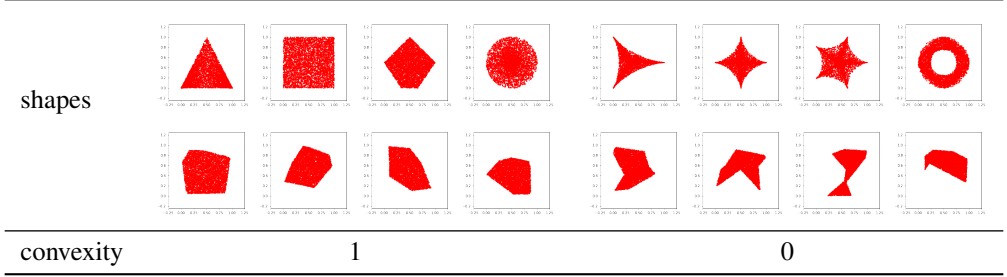

Figure 5: Convexity data set.

**PH pipeline** To build a filtration, we consider cubical complexes filtered by tubular functions that measure the distance of points from a certain line, see Definition 1 for a precise definition. For a good choice of line, multiple components would be seen in the filtration of a point cloud sampled from a concave shape, at least for some values $r \in \mathbb{R}$ (see also the illustrations in Appendix A.1). For this reason we consider the cubical complex, rather than the standard Vietoris-Rips simplicial complex wherein these separate components could be connected with an edge (for details, see Appendix E.1). To build an image from the point cloud, we construct a $20 \times 20$ grid and define a pixel as black if it contains any point-cloud points, and white otherwise.

Since sources of concavity can lie anywhere on the point cloud, we consider nine different lines for the tubular filtration function (for a visualization of the pipeline, see Appendix E.1). For each of the nine lines, we extract 0-dimensional PD, as it captures information about the components. If the point cloud is thus sampled from a convex shape, its PD will only see a single component for any line, whereas there will be multiple components at least for some lines for points clouds sampled from concave shapes. For this reason, for each of the nine lines, we focus our attention only on the lifespan of the second most persisting cycle. We can consider this 9-dimensional vector as our PH signature, but in our experiment choose an even simpler summary: the maximum of these lifespans, since we only care if there are multiple components *for at least one line*. This scalar could even be used as some measure of a level of concavity of a shape.

**Results** As already indicated, to gain some insights into how well the different approaches discriminate convexity from concavity rather than differentiating between the different basic shapes, we look at the classification accuracies under different conditions (Figure 6, with detailed results across multiple runs in Appendix E.2). We start with the easiest case, where both the train and test data consist of the simple regular convex and concave shapes (Figure 5, first row), and then proceed to the scenario where both train and test data are random shapes (Figure 5, second row). Next we proceed to out-of-distribution test data, where we train on the regular and test on random shapes, or vice versa. In every case, we train on 400 and test on 80 point clouds. The results show that PH is able to detect

convexity, surpassing other methods significantly in all scenarios except for PointNet in the scenario on the data set of regular shapes which performs on par. Results reported in Appendix E.3 show PH is also computationally efficient.

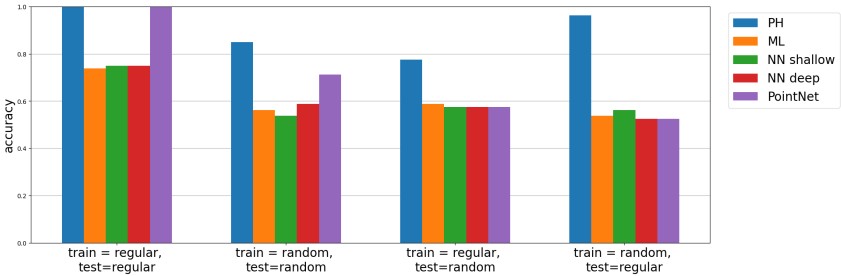

Figure 6: Persistent homology can detect convexity.

The PH pipeline above makes a wrong prediction when concavity is barely pronounced, or if it is missed by the selected tubular filtration lines (for details see Appendix E.4). However, the accuracy of PH can easily be improved simply by considering a finer resolution for the cubical complexes and/or additional tubular filtration lines. The particular PH pipeline summarized in this section would also make a wrong prediction if the data set would include shapes that have small or non-central holes, e.g., a square with a hole in the top left corner. In this case, the accuracy could also be improved by considering a finer cubical complex resolution and by considering additional non-central tubular filtration lines within shapes, or by adding (the maximum lifespan of the) 1-dimensional PH which captures holes. The pipeline is not limited to polygons, or connected shapes, and it can be generalized to surfaces in higher dimensions (Theorem 1). In Appendix G, we also consider the real-world data set FLAVIA for which we demonstrate that a PH pipeline is effective in detecting a continuous measure of convexity.

## 6 Implications for PH interpretation: topology vs. geometry

Here we discuss how our results contribute to the important and ongoing discussion about the interpretation of long versus short persistence intervals. When PH was first introduced in the literature, the long intervals were commonly considered as important or "signal", and short intervals as irrelevant or "noise" [38]. Subsequently the discussion has refined when it was shown that short and medium-length persistence intervals have the most distinguishing power for specific types of applications [8, 99]. The current understanding is roughly that long intervals reflect the topological signal, and (many) short intervals can help in detecting geometric features [3]. We believe that our work brings new insight into this discussion. We give a summary of the implications of our work in this section, and we provide a more detailed discussion in Appendix F.

**Topology and long persistence** Stability results guarantee that a number of longest persistence intervals reflect the topological signal, i.e., the number of cycles [32]. These theorems give information about the threshold that differentiates between long and short persistence intervals. In Section 3, where we focus on the topology of underlying shapes, the experiments demonstrate that this threshold can be learned with simple machine learning techniques. However, it is important to highlight that the distinction between long and short persistence is vague in practice. Indeed, seemingly short persistence intervals capture the topology in Section 3, but the second-longest interval is topological noise in Section 5, since every shape in the data set has only a single component (although this second-longest interval captures important *geometric* information, what enabled us to discriminate between convex and concave shapes). These two problems also clearly indicate how the long intervals that encode topology might or might not be irrelevant, depending on the signal of the particular application domain.

**Geometry and short persistence** The current understanding is that (many) short persistence intervals detect geometry. Section 4 confirms that this indeed can be the case. However, we highlight that *all* cycles can encode geometric information, such as the information about their size (with respect to the Vietoris-Rips and related filtrations, as in Sections 3 and 4) or their position (with

respect to the height or tubular filtration, as in Section 5). This further implies that, depending on the application, *any number* of short or intervals of *any persistence* can be important, which was clearly demonstrated in Section 5, where we show that a single interval detects convexity.

# 7    Conclusions

**Main contribution**    The goal of this work is to gain a better understanding of the topological and geometric features that can be captured with persistent homology. We focus on the detection of number of holes (Section 3), curvature (Section 4), and convexity (Section 5). Theoretical evidence for the first two classes of problems has been established in the literature, and we prove a new result that guarantees that PH can detect convexity (Theorem 1). We also experimentally demonstrate that PH can solve all three problems for synthetic point clouds in $\mathbb{R}^2$ and $\mathbb{R}^3$, outperforming a few baselines. This is true even when there is limited training data and computational resources, and for noisy or out of distribution test data. For convexity detection, we also show the effectiveness of PH in a real world plant morphology application.

**Relevance**    Firstly, the findings point the way to further advances in utilizing the potential of PH in applications: we can expect PH to be successful for classification or regression problems where the data classes differ with respect to number of holes, curvature and/or convexity. Detailed guidelines are discussed in Appendix F. Due to the crucial role of shape classification in understanding and recognizing physical structures and objects, image processing and computer vision [73], our results demonstrate that PH can—to borrow the words from Wigner [112]—"remain valid in future research, and extend, to our pleasure", and lesser bafflement, to a variety of applications. Secondly, the results advance the discussion about the importance of long and short persistence intervals, and their relationship to topology and geometry (Section 6). Topology is captured by the long intervals, geometry is encoded in all persistence intervals, and any interval can encode the signal in the particular application domain.

**Limitations**    The results focus on three selected problems and data sets, and it would therefore be interesting to consider other tasks. In addition, we do not have an extensive comparison of the state-of-the-art for the given problems. Our work seeks to understand if PH is successful for a selected set of tasks by benchmarking it against some well-performing methods.

**Future research**    An in-depth analysis of the hypothetical applications discussed in this paper (Appendix F) and selected success stories of PH from the literature could further improve our understanding of the topological and geometric information encoded in PH, and the interpretation of persistence intervals of different lengths. Alternative approaches for detection of convexity with PH (relying on higher homological dimensions, or multiparameter persistence) are particularly interesting avenues for further work. Furthermore, even though our results imply that PH features are recommended over baseline models for the three selected classes of problems, they also provide inspiration on how to improve existing learning architectures. Further work could investigate deep learning models on PH (and standard) features or kernels [14, 55, 89, 116], an additional network layer for topological signatures, or PH-based priors, regularization or loss functions [14, 34, 36, 37, 56, 113, 121].

**Potential negative societal impact**    While we recognize that the applications of shape analysis can take many different directions, we do not foresee a direct path of this research to negative societal impacts.

**Funding transparency statement**    This research was conducted while RT was a Fulbright Visiting Researcher at UCLA. GM acknowledges support from ERC grant 757983, DFG grant 464109215, and NSF-CAREER grant DMS-2145630. NO acknowledges support from the Royal Society, under grant RGS\R2\212169.

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
