## Appendix

The appendix is organized into the following sections.

- Appendix A Theoretical results
- Appendix B: Experimental details; pipelines, training and hyperparameter tuning
- Appendix C: Additional experimental details for number of holes
- Appendix D: Additional experimental details for curvature
- Appendix E: Additional experimental details for convexity
- Appendix F: Guidelines for persistent homology in applications; long and short persistence intervals
- Appendix G: Persistent homology detects convexity in FLAVIA data set

## A   Theoretical results

In this section, we provide the proof of our Theorem 1 that guarantees that PH can be used to detect convexity. We then formulate Theorem 2 and Theorem 3 from the literature, that summarize known theoretical guarantees that PH can detect the number of holes and curvature. At the end of the section, we also discuss some known results about the theoretical computational complexity of PH.

### A.1   Convexity

In our pipeline for the detection of convexity we consider tubular filtrations, akin to the concept of tubular neighborhoods in differential topology. Given a subspace $X$ of $\mathbb{R}^d$, and a line $\alpha \subset \mathbb{R}^d$, we consider all the points in $X$ that are at a specific distance $r$ from the line. By varying $r$, we then obtain a filtration of $X$. We note that while a tubular neighborhood is defined with respect to any curve, here instead we focus on the special case of (closed) neighborhoods with respect to a line.

**Definition 1** (Tubular filtration). Given a line $\alpha \subset \mathbb{R}^d$, we define the **tubular function with respect to** $\alpha$ as follows:

$$\tau_\alpha \colon \mathbb{R}^d \to \mathbb{R}$$
$$x \quad \mapsto \mathrm{dist}(x, \alpha) ,$$

where $\mathrm{dist}(x, \alpha)$ is the distance of the point $x$ from the line $\alpha$. Given $X \subset \mathbb{R}^d$ and a line $\alpha \subset \mathbb{R}^d$, we are interested in studying the sublevel sets of $\tau_\alpha$, i.e., the subsets of $X$ consisting of points at a specific distance from the line. We define

$$X_{\tau_\alpha, r} = \{x \in X \mid \tau_\alpha(x) \le r\} = \{x \in X \mid \mathrm{dist}(x, \alpha) \le r\} .$$

We call $\{X_{\tau_\alpha, r}\}_{r \in \mathbb{R}_{\ge 0}}$ the **tubular filtration with respect to** $\alpha$.

We first formulate and prove our main theorem below, and then discuss the need for tubular filtration in Remark 2.

**Remark 1** (Different notions of components). In the proof of Theorem 1 we need to work with path-connected components. We note that while in the main part of this manuscript we always only use the term "components", more precisely one would need to distinguish between "connected components" and "path-connected components". For the purposes of the majority of the spaces that we consider in this work, the two notions are equivalent. Thus, we often simply refer to these as "components".

**Theorem 1** (Convexity). Let $X \subset \mathbb{R}^d$ be triangulable[7]. We have that $X$ is convex if and only if for every line $\alpha$ in $\mathbb{R}^d$ the persistence diagram in degree 0 with respect to the tubular filtration $\{X_{\tau_\alpha, r}\}_{r \in \mathbb{R}_{\ge 0}}$ contains exactly one interval.

---

[7]Namely, there exists a simplicial complex $K$ and a homeomorphism $h \colon |K| \to X$ from the geometric realization of $K$ to X.

*Proof.* We first note that the persistence diagram in degree 0 of $\{X_{\tau_\alpha,r}\}_{r\in\mathbb{R}_{\geq 0}}$ exists, since the singular homology in degree 0 (and with coefficients in a field) of $X_{\tau_\alpha,r}$ is finite-dimensional for any $r \in \mathbb{R}_{\geq 0}$; the existence of the persistence diagram then follows from [28, Theorem 2.8].[8]

Assume that $X$ is convex. By definition, we have that for all $p_1, p_2 \in X$ the straight-line segment between $p_1$ and $p_2$ is contained in $X$. Let $\alpha$ be any line in $\mathbb{R}^d$ and $r \in \mathbb{R}$. By elementary properties of Euclidean spaces (similarity of triangles, see Figure 7), we have that if $\text{dist}(p_1, \alpha) \leq r$ and $\text{dist}(p_2, \alpha) \leq r$, then also $\text{dist}(q, \alpha) \leq r$ for any point $q$ on the line segment between $p_1$ and $p_2$ (Figure 7). By the definition of tubular function, this means that $p_1, p_2 \in X_{\tau_\alpha,r}$ implies that $q \in X_{\tau_\alpha,r}$. Therefore the straight-line segment between $p_1$ and $p_2$ is contained in $X_{\tau_\alpha,r}$, which means that $X_{\tau_\alpha,r}$ is convex, and thus path-connected. We therefore have that for any line $\alpha$, the persistence diagram in degree 0 of $\{X_{\tau_\alpha,r}\}_{r\in\mathbb{R}}$ contains a single interval.

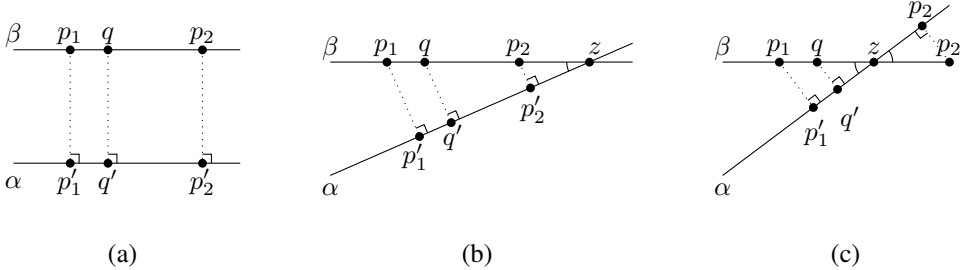

(a)                                        (b)                                        (c)

Figure 7: The distance $\text{dist}(p, \alpha)$ from a point $p$ to a line $\alpha$ is defined as the distance between $p$ and the projection $p'$ of $p$ to line $\alpha$ (denoted with dotted lines in the figure). For any two points $p_1, p_2 \in \mathbb{R}^d$, $q \in \mathbb{R}^d$ on a line segment between $p_1$ and $p_2$, and a line $\alpha$ in $\mathbb{R}^d$, we have that the lines $\beta$ passing through $p_1$ and $p_2$ is either (a) parallel, or (b)-(c) intersects the line $\alpha$ in a point $z = \beta \cap \alpha$. In the former case, per definition of parallel lines, we have that $\text{dist}(p_1, \alpha) = \text{dist}(q, \alpha) = \text{dist}(p_2, \alpha) = \text{dist}(\beta, \alpha)$. In the latter cases, we have a similarity of triangles $\triangle p_1 p_1' z$, $\triangle qq'z$ and $\triangle p_2 p_2' z$, since they all have a 90°angle, and share the angle $\angle(\beta, \alpha)$. Since $\text{dist}(q, s)$ lies between $\text{dist}(p_1, s)$ and $\text{dist}(p_2, s)$, the triangle similarity implies that $\text{dist}(q, \alpha)$ lies between $\text{dist}(p_1, \alpha)$ and $\text{dist}(p_2, \alpha)$.

Assume now that $X$ is concave. Then by definition there exist $p_1, p_2 \in X$ and a point $q$ on the straight-line segment between $p_1$ and $p_2$ such that $q \notin X$. Since $X$ is closed, we have that there exists $\epsilon > 0$ such that $B(q, \epsilon) \subset \mathbb{R}^d \setminus X$ (Figure 8). Let $\alpha$ be the line passing through $p_1$ and $p_2$, and let $0 \leq r \leq \epsilon$. We then have that $\text{dist}(p_1, \alpha) = \text{dist}(p_2, \alpha) = 0 \leq r$, so that $p_1, p_2 \in X_{\tau_\alpha,r}$. We claim that the subset $X_{\tau_\alpha,r}$ is not path-connected.

Let us assume otherwise, i.e., that that $X_{\tau_\alpha,r}$ is path-connected. Equivalently, there is a path connecting $p_1$ and $p_2$ and which is contained in $X_{\tau_\alpha,r}$. Then such a path would have to intersect the hyperplane $\beta$ passing through $q$ and orthogonal to line $\alpha$. To see why, we first note that the complement $\mathbb{R}^d \setminus \beta$ of the hyperplane is disconnected with two connected components, each containing one of the two points $p_1$ and $p_2$. If the path would not intersect the hyperplane, it would be contained in the complement of the hyperplane, but not entirely contained in one of the connected components, which yields a contradiction to the path being connected. By construction, this point of intersection $z \in X$ lies on the path between $p_1$ and $p_2$, and therefore $z \in X_{\tau_\alpha,r}$, or equivalently, $\tau_\alpha(z) = \text{dist}(z, \alpha) \leq r$. Since $z$ is also contained in the hyperplane $\beta$ orthogonal to the line $\alpha$ and passing through $q$, we have that $\text{dist}(z, q) = \text{dist}(z, \alpha) \leq r$, i.e., $z \in B(q, r) \subset \mathbb{R}^d \setminus X$, which is a contradiction to $z \in X$. Therefore, for any $0 \leq r \leq \epsilon$, the set $X_{\tau_\alpha,r}$ is not path-connected, so that the 0-dimensional PD on the tubular filtration with respect to $\alpha$ contains at least two intervals.

$\square$

**Remark 2** (Relationship with the height filtration). To illustrate the need for the tubular filtration, we discuss how it compares to the height filtration that is well-established in the literature. For a given

---

[8]We note that while in this proof we need to consider singular homology, when computing PH in practice one works with either simplicial or cubical homology. For the types of spaces that we consider in our work, all homology theories are equivalent. See [60] for a discussion of the equivalence between simplicial and cubical homology, and [53, Chp. 2] for the equivalence of singular and simplicial homology.

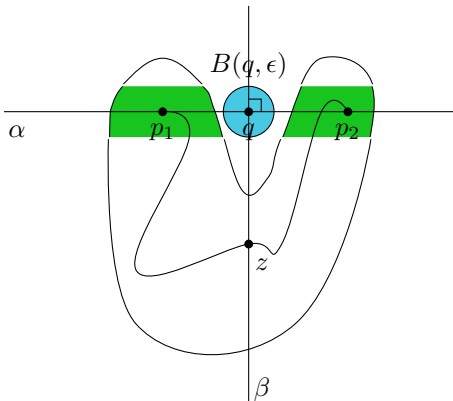

Figure 8: For a concave shape $X$, there exists a tubular filtration line $\alpha$ so that the resulting 0-dimensional PD sees multiple path-connected components (in green). Note that the path in the figure can exist, but it cannot lie completely in the particular sublevel set (in green).

shape $X \in \mathbb{R}^d$ and a unit vector $v \in S^{d-1}$, the height function $\eta_v : \mathbb{R}^d \to \mathbb{R}$ is defined via the scalar product, $\eta_v(x) = x \cdot v$. If we consider the hyperplane that is orthogonal to the vector $v$ and passes through the origin $(0, 0, \ldots, 0) \in \mathbb{R}^d$, the sublevel set $X_{\eta_v, r}$ corresponds to the area in $X$ above or in the hyperplane and below or at height $r \in \mathbb{R}$, and the complete area of $X$ below the hyperplane (where the scalar product is negative) (Figure 9, column 1). Note that it is possible to recenter the shape at the origin, so that the hyperplane does not need to pass through the origin.

We note that 0-dimensional PH with respect to the height filtration can help us detect *some* concavities in $\mathbb{R}^d$, what is the case for panel (a) in Figure 9, where we clearly see multiple path-connected components in green. However, for shapes in $\mathbb{R}^2$ where a source of concavity is a hole within the shape, such as the annulus-like shape in row 2 of Figure 9, the sublevel sets with respect to the height filtration will only see a single path-connected component, see panel (d) in Figure 9. Indeed, there is no unit vector $v$ for which the sublevel set $X_{\eta_v, r} = \{x \in X \mid \eta_v(x) = x \cdot v \leq r\}$ contains more than one path-connected component. If we adjust the definition of the filtration function and let $\eta'_v(x) = |x \cdot v|$, the sublevel set $X_{\eta'_v, r}$ corresponds to the area in $X$ above and below the hyperplane, and within height $r \in \mathbb{R}$ in both directions. In other words, $X_{\eta'_v, r}$ is the area of $X$ within a given distance from the hyperplane. 0-dimensional PH with respect to the absolute height function $\eta'_v$ enables us to detect any concavity in $\mathbb{R}^2$, see (b) and (e) in Figure 9, since the sublevel set consists of multiple path-connected components (in green). In $\mathbb{R}^2$, the tubular filtration corresponds to the absolute height filtration. However, neither the height nor the absolute height filtration can detect concavity in higher dimensions. Indeed, consider a sphere-like shape as an example concave shape in $\mathbb{R}^3$ (Figure 9, row 3). The sublevel sets $X_{\eta_v, r}$ and $X_{\eta'_v, r}$ will result in a single path-connected component, see the green areas respectively in panels (g) and (h) in Figure 9. On the other hand, the area within a distance from some *line* (points in $X$ that are within a given tube) can result in two (disconnected) disks on polar opposites on the sphere, see panel (i) in Figure 9.

However, we note that there are likely alternative approaches that can rely on PH with respect to the (absolute) height filtration to detect concavity. One possibility would be to also consider homology in higher dimensions (although, we note that 0-dimensional PH is faster to compute, see Appendix A.4). For shapes in $\mathbb{R}^2$, it is sufficient to consider the 0- and 1-dimensional PH with respect to the height filtration. Indeed, although the annulus-like shape (Figure 9, row 2) does not see multiple path components with respect to any height filtration function, there is a 1-dimensional hole which points to a concavity. This, however, does not generalize to higher dimensions: Indeed, a sublevel set of a sphere with respect to any height filtration will result in a spherical cap which has trivial homology in dimensions 0, 1 and 2, see panel (g) in Figure 9. On the other hand, the sublevel set of the absolute height function on the sphere will always consist of a single path-connected component, but we can capture 1-dimensional holes, see panel (h) in Figure 9. Another interesting example of a concave surface in $\mathbb{R}^3$ is a ball with a dent on the north pole, i.e., a crater. This concavity cannot be detected with 0-dimensional PH with respect to any (absolute) height filtration. For the example surface, an interesting direction would be looking at the surface "from the top" (horizontal hyperplane), but PH

would start by seeing a circle, and then the crater itself — always a single path-connected component. However, 1-dimensional PH with respect to this filtration would capture a hole, implying that the surface is concave.

Another possibility could be to study (the computable) multiparameter 0-dimensional PH by scanning shapes from multiple directions *simultaneously*. For the sphere, 0-dimensional PH would capture the two path-connected components with the bi-filtration that looks at the shape with respect to the horizontal hyperplane denoted in the panel (h) in Figure 9, and the orthogonal hyperplane passing through the shape. We note that while the theory and computations for multiparameter persistence are hard, there have been some recent advances, see, e.g., [101]. This is similar to slicing the shape with a hyperplane, and then studying the single-parameter 0-dimensional PH of the slice. The alternative approaches that we briefly discuss here are an interesting avenue for further work.

| height | absolute height | tubular |
|---|---|---|
| scalar product $\eta_v(x) = x \cdot v$ | distance from hyperplane $\eta'_v(x) = |x \cdot v| = \text{dist}(x, \alpha)$ | distance from line $\tau_\alpha(x) = \text{dist}(x, \alpha)$ |

Figure 9: Sublevel sets of the height (column 1), absolute height (column 2) and tubular function (column 3) for three example concave shapes in $\mathbb{R}^2$ and $\mathbb{R}^3$. Concavity is detected with multiple 0-dimensional persistence intervals, which reflect the multiple path-connected components in the filtration. The sublevel set (in green) has multiple path-connected components for the height function only for the shape in row 1, for the absolute height only for shapes in rows 1 and 2, whereas for the tubular function this is the case for any concave shape in $\mathbb{R}^d$.

**Remark 3** (Relationship with the Persistent Homology Transform). There is a lot of work done on studying the so-called Persistent Homology Transform (PHT), which is given by considering the PH on the height filtration with respect to all unit vectors [41, 49, 104]. Such a topological summary that has been shown to be a sufficient statistics for probability densities on the space of triangulizable subspaces of $\mathbb{R}^2$ and $\mathbb{R}^3$, respectively [104].

For practical purposes it would not be feasible to have to consider all unit vectors in the PHT. Luckily, there are known results on the sufficient number of directions [41]. In computational experiments in [104] on the MPEG-7 silhouette database of simulated shapes in $\mathbb{R}^2$ [67, 96] and point clouds in $\mathbb{R}^3$ obtained from micro-CT scans of heel bones [13], the PHT is approximated by looking respectively at 64 evenly spaced directions and 162 directions constructed by subdividing an icosahedron. Furthermore, 0- and/or 1-dimensional PH with respect to the height and/or related

radial filtration has also been used as a 3-dimensional shape descriptor in [22], for analysis of brain artery trees [8], or classification of MNIST images of handwritten digits [48].

We note that the theoretical results related to the Persistent Homology Transform focus on a complete description of shapes, whereas here we are interested in investigating to what extent PH can detect convexity and concavity.

**Remark 4** (Detection of convexity with PH in practice). To calculate PH in practice, the sublevel sets $K_r$ need to be approximated with simplicial or cubical complexes (see Section 2.1). The PH pipeline in our computational experiments for convexity detection (Section 5) relies on cubical complexes, but it is possible to do so also with the Vietoris-Rips complex relying on geodesic distances. For details, see Appendix E.1, where we conclude that it is important to ensure that the singular homology of each of the sublevel sets is properly reflected with the homology of the complex. For convexity detection, this means that the complexes of concave shapes should also see multiple connected components (that do not connect with a simplex).

Furthermore, we note that, to simply differentiate between convex and concave shapes (binary classification problem), it would be sufficient to only consider 0-dimensional *homology* of the intersection of $X$ with the line $\alpha$, i.e.,

$$X_{\tau_\alpha,0} = \{x \in X \mid \tau_\alpha(x) = \text{dist}(x, \alpha) \leq 0\} = X \cap \alpha,$$

which is easier to compute than the multi-scale PH. This intersection is a line segment for convex shapes, and for concave shapes it is a union of disconnected segments on a line. Therefore, convex shapes will have $\beta_0(X \cap \alpha) = 1$, and concave shapes will have $\beta_0(X \cap \alpha) > 0$ (one could think of this as persistence intervals $[0, +\infty)$ that all have the same lifespan). However, in practical applications, it is often useful to capture a more detailed information about concavity. For example, for the plant morphology application we consider in Appendix G, the goal is to capture a continuous measure of concavity (regression problem). Then, the (different) lifespans of the second (or also third, fourth, ...) most-persisting connected component can provide important additional information. Moreover, in applications $X$ is typically finite, e.g., a point cloud, so that one would still need to approximate $X \cap \alpha$ (points on a line segment) with a complex in order to calculate the homology (see paragraph above). In other words, we would need to choose an appropriate scale $r \in \mathbb{R}$ that would ensure that the complex faithfully reflects the homology of $X \cap \alpha$, which is a non-trivial task.

## A.2 Number of holes

In our pipeline to detect the number of holes, we use the alpha complex, for which several theoretical guarantees have been proven.

The Nerve Lemma (see, e.g., [44]) guarantees that the alpha complex of a set of points has the same homology-type as the space obtained by taking unions of balls of a certain radius centered around the points. Whether this union of balls has the same homology-type as the space from which the points are sampled depends on properties of the sample. If the sample is dense enough, then it has been shown that, for a suitable value of the scaling parameter, the alpha complex has the same number of holes as the original space, for instance under the assumption on the space being a smooth manifold [77]. For ease of reference, we reproduce here the result from [77].

**Theorem 2** (Number of holes). Let $M$ be a compact smooth manifold, and $X$ a set of points sampled uniformly at random from $M$. Then there exists $r \in \mathbb{R}$ such that the homology of the alpha simplicial complex $\alpha(X, r)$ is isomorphic to the singular homology of the underlying manifold $M$.

*Proof.* [77, Theorem 3.1] implies that there exists $r \in \mathbb{R}$ such that the singular homology of the $\cup_{x \in X} B(x, r)$ is isomorphic to the singular homology of $M$. By the Nerve Lemma we then know that the simplicial homology of the alpha complex $\alpha(X, r)$ is isomorphic to the singular homology of $\cup_{x \in X} B(x, r)$. $\square$

The alpha complex is known to approximate the Vietoris–Rips complex, in the sense that the respective persistence modules are interleaved, see, e.g. [44].

## A.3 Curvature

Here we reproduce the theoretical guarantee provided in [17].

**Theorem 3** (Curvature). Let $M$ be a manifold with constant curvature $\kappa$, and $D_\kappa$ be a unit disk on $M$. Let further $X$ be a point cloud sampled from $X$, according to the probability measure proportional to the surface area measure. Then, PH of $X$ recovers $\kappa$.

*Proof.* Given $\kappa$, [17, Theorem 14] establishes an analytic expression for the persistence ($p = d/b$) of triangles to the curvature $\kappa$ of the underlying manifold. This function is continuous and increasing, so that persistence recovers curvature. $\qquad\square$

### A.4 Computational complexity

In this section we discuss how our pipelines are affected by the size $n$ of the point cloud and the dimension $d$ of the embedding space.

There exist several efficient algorithms for the computation of PH, many coming with heuristic guarantees on speed-ups for the computation (see survey [80] for an overview, and references therein). For the purposes of this discussion, we will focus on the standard algorithm, which has a computational complexity which is cubical in the number $N$ of simplices contained in the filtered complex [123], i.e., the computational complexity is $\mathcal{O}(N^3)$. Thus, to better understand how our pipelines generalize to higher-dimensional point clouds, in the following we explain how the sizes $N$ of the different types of complexes that we consider are affected by the size $n$ and dimension $d$ of the point cloud.

**Number of holes**    For the detection of the number of holes, in our experiments we relied on alpha simplicial complex. In the worst case, the size $N$ of the alpha complex is $\mathcal{O}\left(n^{\lceil d/2 \rceil}\right)$.

**Curvature**    We used the Vietoris-Rips simplicial complex for curvature detection. The size $N$ of the Vietoris–Rips complex is $\mathcal{O}\left(n^{k+2}\right)$, where $k$ is the maximum PH degree that we are interested in computing.

In general, the choice between the alpha and Vietoris-Rips simplicial complex therefore depends on the point cloud size $n$ and dimension $d$ (which are also related, since typically exponentially more points are needed to properly sample a shape in higher dimensions), and the highest homological dimension $k$ that is of interest for the given problem at hand. For point clouds with a given number $n$ of points, the alpha complex is better suited in lower dimensions ($d = 2, 3$), and provided that the point cloud is embedded in Euclidean space.

**Convexity**    To detect convexity, we relied on cubical complexes. For data sets that have an inherently cubical structure, using cubical complexes may yield significant improvements in both memory and runtime efficiency [109]. This is particularly true for high dimensional data, since the ratio between the size of the Vietoris-Rips simplicial complex compared to a cubical complex is exponential in dimension $d$ [33].

In our construction, given a point cloud in $\mathbb{R}^d$ and a fixed $c \in \mathbb{N}$, we bin the points into $c^d$ cubes of the same volume. In the worst case, the size $N$ of the cubical complex of the resulting structure is $\mathcal{O}(3^d c^d)$ (and thus does not depend on the number $n$ of point cloud points).

In addition, it is important to note that our convexity detection pipelines only uses the 0-dimensional persistent homology, which has a reduced complexity since one only needs to construct the complex up to dimension 1. It is fairly easy to compute 0-dimensional PH in near-linear time with respect to the number $N$ of simplices by using union-find data structures [43, 44, 109]. For this reason, it is an important advantage of our pipeline that it only relies on 0-dimensional PH, without needing to calculate PH in higher homological dimensions.

Detecting convexity, however, poses additional challenges. Testing convexity is fundamentally a hard problem in high dimensions, related to the hardness of computing convex hulls in high dimensions, and unfortunately we cannot hope for free lunch. In our PH convexity detection pipeline, unlike for the detection of the number of holes or curvature, we calculate PH across multiple tubular-filtration lines, whose number also grows with the dimension $d$ since sufficiently many filtrations need to be considered (and the same would be the case - we would have to consider multiple tubular directions, if we considered simplicial instead of cubical complexes, see Figure 20). This could be circumvented by considering (quasi-)random lines. To conclude, specifying a desired computation budget and number of filtrations in advance (leading to a corresponding accuracy tradeoff), our PH pipeline can

be used to obtain fast estimates of convexity. It can also be used to compute a continuous measure of convexity (as we demonstrate on the real-world FLAVIA data set of leaf images in Appendix G), or convexity at a given resolution, depending on the resolution of the filtration, which in some cases may be more useful than the binary label (convex or concave).

# B    Experimental details

## B.1    Reproducibility and computer infrastructure

The data and code developed for this research are made publicly available at `https://github.com/renata-turkes/turkevs2022on`. All our computations were conducted using a 2.7Ghz vCPU core from a DGX-1 + DGX-2 station.

## B.2    Hyperparameter tuning and training procedure for the individual pipelines

In this section, we provide more details about the pipelines that were compared in the computational experiments:

- SVM on persistent homology features (PH),
- simple machine learning (ML) baseline - SVM on distance matrices,
- fully connected neural network (NN) on distance matrices, and
- PointNet (PointNet) on raw point clouds.

For each pipeline, we list the hyperparameters that were tuned. To ensure a fair comparison of the different approaches, we used the same train and test data across all the pipelines. We used `sklearn GridSearchCV` based on cross validation with 3 folds and random splits, and returned the hyperparameter values that resulted in the highest accuracy for classification problems (Section 3 and Section 5), or the lowest mean squared error for regression problems (Section 4). We also list relevant software and licenses.

**PH**    The general steps to extract PH features are visualized in Figure 10. To calculate PH in Section 3 and Section 5 we use `GUDHI` [100], and in Section 4 we use `Ripser` [7, 102], which are persistent homology libraries in Python, available under the MIT (GPL v3) license. For the DTM filtration in Section 3, we choose $m = 0.03$, so that $0.03 \times 1\,000 = 30$ nearest neighbors are used to calculate the filtration function. Grid search is performed to choose the best persistence signature and classifier or regressor as described below.

- In Section 3 and Section 4, we select between:
  - simple signature of 10 longest lifespans,
  - persistence images with resolution $10 \times 10$, bandwidth $\sigma \in \{0.1, 0.5, 1, 10\}$, and weight function $\omega(x, y) \in \{1, y, y^2\}$, and
  - persistence landscapes with resolution of 100, and considering the longest 1, 10 or all persistence intervals.
- We use SVM (`sklearn SVC` and `SVR` for classification and regression respectively) on the PH signature, with the regularization parameter $C \in \{0.001, 1, 100\}$. The latter tunes the trade off between correct classification of training data and maximization of the decision function's margin.

**ML**    In our experiment, the input for the simple machine learning (ML) pipeline is the normalized matrix of pairwise distances between point-cloud points. For a given point cloud $X = \{x_1, \ldots, x_n\} \subset \mathbb{R}^d$, the corresponding distance matrix is the $n \times n$ matrix $D \in \mathbb{R}^{n \times n}$, with entries $D_{ij} = \text{dist}(x_i, x_j)$ corresponding to the Euclidean distance for the detection of the number of holes (Section 3) or convexity (Section 5), and hyperbolic, Euclidean or spherical distance for curvature detection (Section 4). We take the entries above the diagonal flattened into a vector. Since the dimension of distance matrices scales with the square of the number of points, we work with subsamples of 100 distinct random points from each point cloud. Similarly as above, we use cross validation to choose the SVM regularization parameter among $C \in \{0.001, 1, 100\}$.

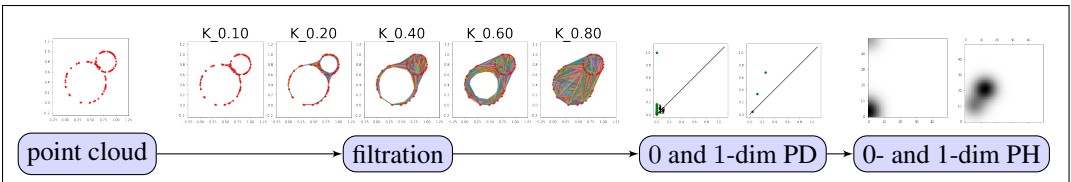

Figure 10: Persistent homology features. To calculate PH for the given point cloud in $\mathbb{R}^2$, we first construct a filtration $\{K_r\}_{r \in \mathbb{R}}$ which approximates $X$ at different scales $r \in \mathbb{R}$,, where $K_r$ is the Vietoris-Rips simplicial complex. 0-dimensional PD has one persistent cycle, reflecting the single component, and a number of short cycles that correspond to the individual point-cloud points that are connected to other ones early in the filtration. 1-dimensional PD summarizes the two holes, whose birth and death values respectively reflect the sparsity along the hole and the size of the hole, as these are the scales $r \in \mathbb{R}$ at which the hole appears and when it is filled in within the filtration. PDs are then represented by PIs, which are vector summaries that can be used in statistical learning frameworks, but many other signatures (denoted, in general, with PH) can be used.

We note that while a distance matrix can be taken as input to a classifier, it depends on the particular and arbitrary labeling of the points in the point cloud and hence it does not account for the label symmetry of point clouds.

**NN**    The normalized distance matrices are also fed to the multi-layer perceptrons (MLPs). We consider the following hyperparameters:

- depth in $\{1, 2, 3, 4, 5\}$ (only for NN deep),
- layer widths in $\{64, 256, 1\,024, 4\,096\}$,
- learning rate in $\{0.01, 0.001\}$,

that are selected through a grid search, with each parameter setting trained for 2 epochs. We use a softmax activation function, and cross entropy and mean squared error as loss functions for classification (Section 3 and Section 5) and regression (Section 4) problems, respectively. Batch normalization (with zero momentum) and a drop out (with a rate of 0.5) are applied after every (input or hidden) layer.

**PointNet**    PointNet [88] is a neural network that takes point clouds as inputs, and is inspired by the invariance of point clouds to permutations and transformations. It incorporates fully-connected MLPs to approximate classification functions, and convolutional layers to capture geometric relationships between features.

In our experiments, we rely on the PointNet model from `keras` [1] under Apache License 2.0. This model implements the architecture from the original PointNet paper [88], which is supplemented with a publicly available code [87], licensed under MIT. We use grid search to tune:

- number of filters in $\{32, 64\}$,
- learning rate in $\{0.01, 0.001\}$.

For each of the problems we consider, the model is trained from scratch using the training set described in the corresponding section. Unlike in the original paper, we do not augment the data during training by randomly rotating the object or jittering position of each point by a Gaussian noise, in order to ensure a fair comparison with the other pipelines.

## C    Additional experimental details for number of holes

### C.1    Data transformations

To test the noise robustness of the different pipelines, in Section 3 we consider the test data consisting of the original point clouds, or point clouds under different transformations (Figure 2). A detailed description of the data transformations is given in Table 1, and the transformations are visualized on

an example point cloud in Figure 11. To define reasonable values for the data transformations, we took inspiration from the affNIST[9] data set of MNIST images under affine transformations.

Table 1: Data transformations.

| Transformation | Definition of transformation |
|---|---|
| rotation | Clockwise rotation by an angle chosen uniformly from $[-20, 20]$ degrees clockwise. |
| translation | Translation by random numbers chosen from $[-1, 1]$ for each direction. |
| stretch | Scale by a factor chosen uniformly from $[0.8, 1.2]$ in the $x$-direction. The other coordinates remain unchanged, so that the point cloud is stretched. Stretching factor of $0.8$ results in shrinking the point cloud by $20\%$, and the factor of $1.2$ makes it $20\%$ larger. |
| shear | Shear by a factor chosen uniformly from $[-0.2, 0.2]$. A shearing factor of $1$ means that a horizontal line turns into a line at $45$ degrees. |
| Gaussian noise | Random noise drawn from normal distribution $\mathcal{N}(0, \sigma)$ with the standard deviation $\sigma$ uniformly chosen from $[0, 0.1]$ is added to the point cloud. |
| outliers | A percentage, chosen uniformly from $[0, 0.1]$, of point cloud points are replaced with points sampled from a uniform distribution within the range of the point cloud. |

Figure 11: An example point cloud under the considered transformations.

## C.2  Pipeline

Figure 12 visualizes the PH pipeline. To reduce the computation times, we approximate point clouds at each scale with the alpha simplicial complex (discussed in Section 2, and in particular, in Section 2.1). The DTM filtration on the point-cloud points is defined as the average distance from a number of nearest neighbors. Therefore, outliers appear only late in the filtration, so that their influence is smoothed out to a great extent. For the example point cloud in the figure, the 1-dimensional PD consists of four persistence intervals with non-negligible persistence or lifespan (PD points far from diagonal) which correspond to the four big holes, and many short persistence intervals that correspond to holes that are seen at some scales due to noise. This is clearly reflected in the vector of sorted lifespans of the 10 most persisting cycles.

## C.3  Performance across multiple runs

Table 2 provides a detailed overview of the results for the detection of the number of holes, when the experiment is repeated multiple times. The accuracy for PointNet varies for different runs, but in any case, we can clearly see that PH performs the best for each individual run. Note also that the performance of ML, NN shallow and NN deep does not drop under affine transformations, since they take the normalized distance matrices as input.

## C.4  Training curves

Figure 13 shows the training curves for the NN and PointNet pipelines. The training set performance of MLPs (shallow and deep) continues improving over epochs, but the validation set performance

---

[9] https://www.cs.toronto.edu/~tijmen/affNIST/

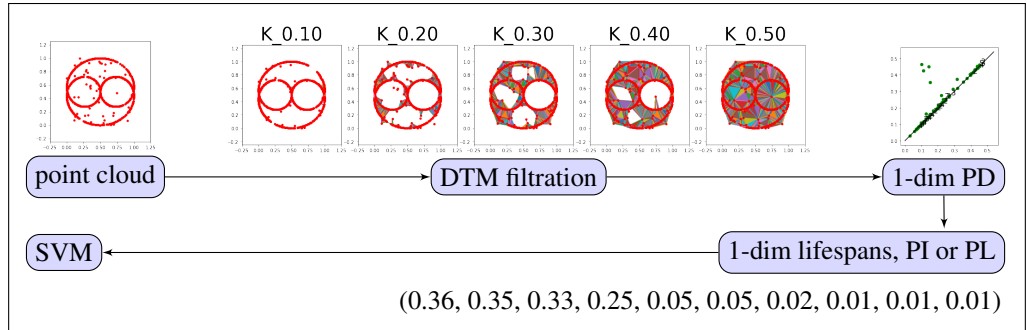

Figure 12: Persistent homology pipeline to detect the number of holes.

quickly saturates and stops improving after a few epochs. PointNet performs well on this task, already after a short number of training epochs. We do not include training curves for the PH and ML pipelines, as these are based on SVMs.

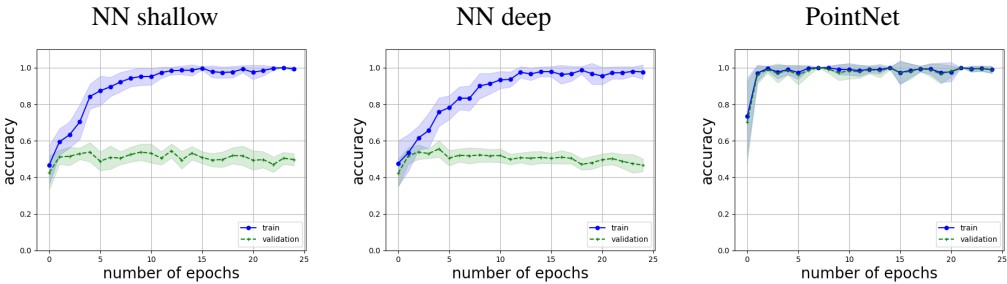

Figure 13: Training curves for the detection of the number of holes.

## C.5 Learning curves

Figure 14 shows the learning curves for every pipeline; i.e., the test accuracy of the trained pipelines depending on the total amount of training data. This serves to evaluate the data efficiency of the different methods. The PH-approaches perform well even for a small number of training point clouds. PointNet also has good performance, although it requires more training data. The other approaches (NN shallow, NN deep, and ML) have poor performance, which does not improve when more training data is available. An explanation for this is that these methods are based on distance matrices and hence cannot directly take advantage of the permutation symmetry of point clouds.

## C.6 Computational resources

Figure 15 visualizes the computational efficiency and memory usage. We see that PH pipeline also performs better with respect to these criteria in comparison to the other methods. The hyperparameter tuning of the PH pipeline does take time (as we consider a wide range of parameters for the different persistence signatures), but Figure 2 shows that even PH simple, where the SVM is used directly on the lifespans of the 10 most persisting cycles (without any tuning of PH-related parameters) performs well.

We note that the difference in the memory usage for data comes from the different types of input that are considered by different pipelines: PDs (lists of persistence intervals) for PH simple and PH, $100 \times 100$ distance matrices for ML and NNs, and $1000 \times 3$ point clouds for PointNet (Appendix B.2).

Table 2: Accuracy across multiple runs for the detection of the number of holes.

| Transformation | Run | PH simple | PH | ML | NN shallow | NN deep | PointNet |
|---|---|---|---|---|---|---|---|
| original | run 1 | 0.94 | **1.00** | 0.67 | 0.52 | 0.50 | **1.00** |
| | run 2 | 0.94 | **1.00** | 0.67 | 0.51 | 0.50 | **1.00** |
| | run 3 | 0.94 | **1.00** | 0.67 | 0.56 | 0.50 | **1.00** |
| | mean | 0.94 | **1.00** | 0.67 | 0.53 | 0.50 | **1.00** |
| | std dev | 0.00 | 0.00 | 0.00 | 0.03 | 0.00 | 0.00 |
| translation | run 1 | 0.94 | **1.00** | 0.67 | 0.52 | 0.50 | 0.23 |
| | run 2 | 0.94 | **1.00** | 0.67 | 0.51 | 0.50 | 0.17 |
| | run 3 | 0.94 | **1.00** | 0.67 | 0.56 | 0.50 | 0.21 |
| | mean | 0.94 | **1.00** | 0.67 | 0.53 | 0.50 | 0.20 |
| | std dev | 0.00 | 0.00 | 0.00 | 0.03 | 0.00 | 0.03 |
| rotation | run 1 | 0.94 | **1.00** | 0.67 | 0.52 | 0.50 | 0.86 |
| | run 2 | 0.94 | **1.00** | 0.67 | 0.51 | 0.50 | 0.57 |
| | run 3 | 0.94 | **1.00** | 0.67 | 0.56 | 0.50 | 0.78 |
| | mean | 0.94 | **1.00** | 0.67 | 0.53 | 0.50 | 0.74 |
| | std dev | 0.00 | 0.00 | 0.00 | 0.03 | 0.00 | 0.15 |
| stretch | run 1 | 0.97 | **0.98** | 0.64 | 0.49 | 0.47 | 0.85 |
| | run 2 | 0.97 | **0.98** | 0.64 | 0.48 | 0.45 | 0.70 |
| | run 3 | 0.97 | **0.98** | 0.64 | 0.52 | 0.51 | **0.98** |
| | mean | 0.94 | **0.98** | 0.64 | 0.50 | 0.48 | 0.84 |
| | std dev | 0.00 | 0.00 | 0.00 | 0.02 | 0.03 | 0.14 |
| shear | run 1 | 0.95 | **1.00** | 0.66 | 0.54 | 0.51 | 0.94 |
| | run 2 | 0.95 | **1.00** | 0.66 | 0.51 | 0.50 | 0.72 |
| | run 3 | 0.95 | **1.00** | 0.66 | 0.56 | 0.51 | 0.96 |
| | mean | 0.95 | **1.00** | 0.66 | 0.54 | 0.51 | 0.87 |
| | std dev | 0.00 | 0.00 | 0.00 | 0.02 | 0.01 | 0.13 |
| Gaussian | run 1 | 0.94 | **1.00** | 0.68 | 0.54 | 0.50 | **1.00** |
| | run 2 | 0.94 | **1.00** | 0.68 | 0.51 | 0.50 | **1.00** |
| | run 3 | 0.94 | **1.00** | 0.68 | 0.56 | 0.51 | **1.00** |
| | mean | 0.94 | **1.00** | 0.68 | 0.54 | 0.50 | **1.00** |
| | std dev | 0.00 | 0.00 | 0.00 | 0.02 | 0.01 | 0.00 |
| outliers | run 1 | 0.82 | **0.93** | 0.62 | 0.55 | 0.49 | 0.70 |
| | run 2 | 0.82 | **0.93** | 0.62 | 0.51 | 0.50 | 0.51 |
| | run 3 | 0.82 | **0.93** | 0.62 | 0.54 | 0.41 | 0.44 |
| | mean | 0.82 | **0.93** | 0.62 | 0.53 | 0.47 | 0.55 |
| | std dev | 0.00 | 0.00 | 0.00 | 0.02 | 0.05 | 0.13 |

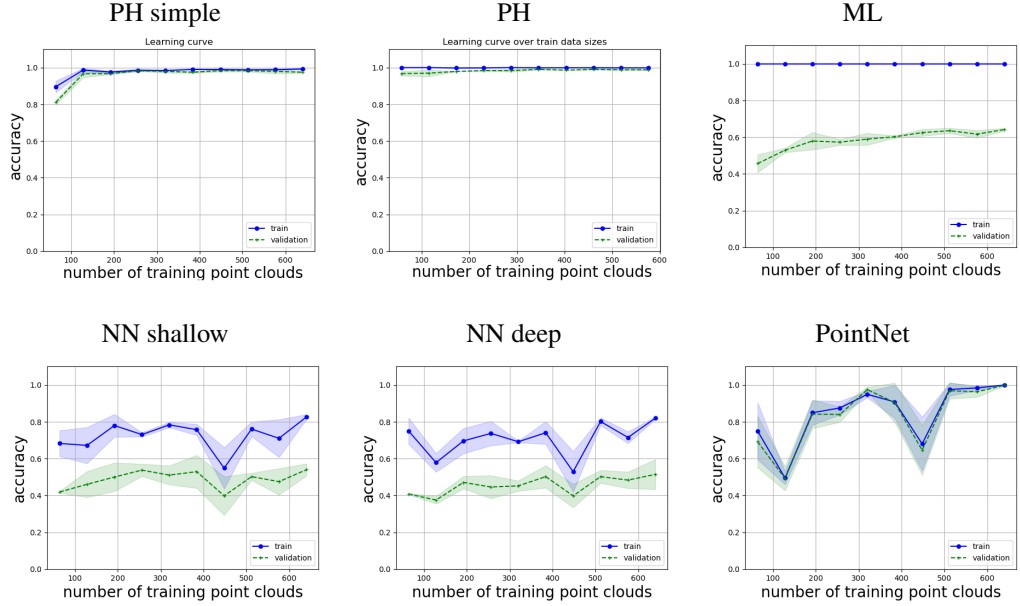

Figure 14: Learning curves for the detection of the number of holes.

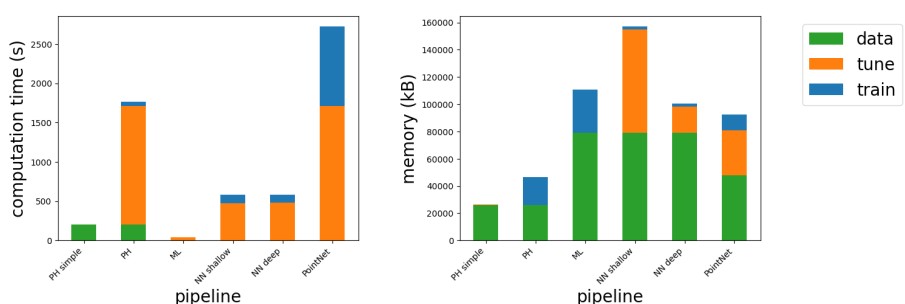

Figure 15: Computational resources for the detection of the number of holes.

# D Additional experimental details for curvature

## D.1 Pipeline

Before we visualize the PH pipeline, we give an illustrative figure that provides some intuition on why PH can detect curvature (Figure 16).

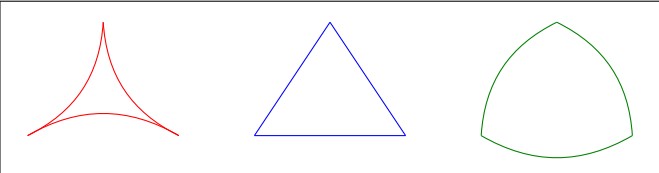

Figure 16: Intuition behind curvature detection with PH. For triangles embedded on a manifold with constant negative (left, in red), zero (middle, in blue) and positive (right, in green) curvature, the length of triangle edges clearly reflect the underlying curvature. Since persistence captures the length of these edges (when the triangle vertices merge into a component), PH can be used to detect curvature.

The PH pipeline to detect curvature (Section 4) is visualized in Figure 17. The example point cloud shown in the figure is in the Euclidean plane. We start by calculating the Euclidean distance matrix, and then construct the Vietoris-Rips filtration from these distances, which approximates the point cloud at different scales. 0-dimensional PD registers one persisting cycle reflecting the single component of the disk (which we ignore, since it is shared by every disk in the data and thus does not contribute to the classification), and many other components which have a short lifespan as they get connected to other point-cloud points early in the filtration. There are no persistent 1-dimensional holes since disks are contractible, but there are many holes with short persistence. PDs are then transformed to a vector summary such as a PI.

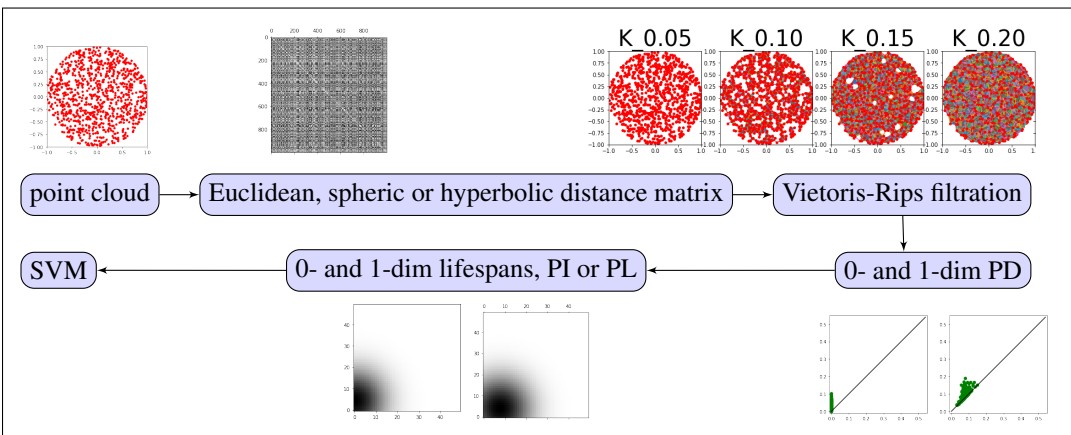

Figure 17: Persistent homology pipeline to detect curvature.

## D.2 Performance across multiple runs

Table 3 shows that (0-dimensional) PH outperforms the other machine- and deep-learning approaches for curvature detection, across multiple experimental runs.

## D.3 Computational resources

Figure 18 visualizes the computational time and memory usage of the different pipelines for this task. The superior performance of the PH pipelines in comparison to other methods (Figure 4) can come at a high cost with respect to the usage of computational resources. However, the simple 0-dim PH pipeline (that only focuses on the lifespans of the PH cycles), which achieves the best predictive power (Figure 4), is the most efficient.

Table 3: Mean squared error across multiple runs for curvature detection.

| Run | 0-dim PH simple | 0-dim PH simple 10 | 0-dim PH | 1-dim PH simple | 1-dim PH simple 10 | 1-dim PH | ML | NN shallow | NN deep | PointNet |
|---|---|---|---|---|---|---|---|---|---|---|
| run 1 | **0.06** | 0.21 | 0.08 | 0.34 | 0.29 | 0.18 | 0.34 | 0.42 | 0.46 | 12.28 |
| run 2 | **0.06** | 0.21 | 0.08 | 0.34 | 0.29 | 0.18 | 0.34 | 0.43 | 0.46 | 0.25 |
| run 3 | **0.06** | 0.21 | 0.08 | 0.34 | 0.29 | 0.18 | 0.34 | 0.66 | 0.43 | 578.28 |
| mean | **0.06** | 0.21 | 0.08 | 0.34 | 0.29 | 0.18 | 0.34 | 0.50 | 0.45 | 196.94 |
| std dev | 0.00 | 0.00 | 0.00 | 0.00 | 0.00 | 0.00 | 0.00 | 0.14 | 0.02 | 330.31 |

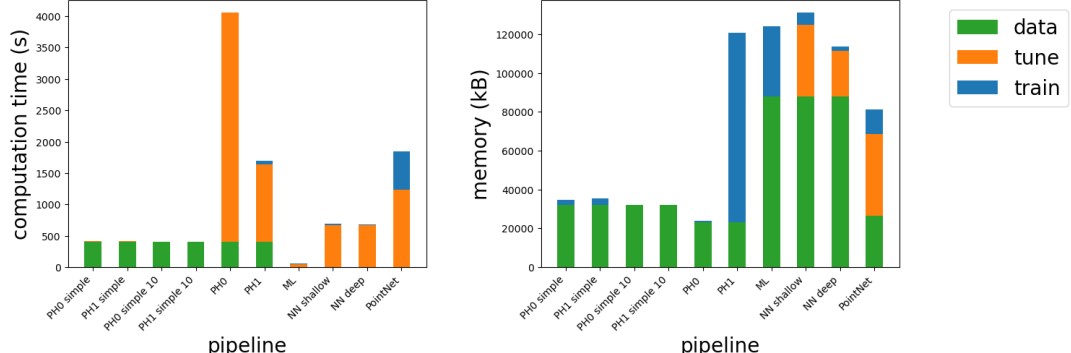

Figure 18: Computational resources for the detection of curvature.

# E    Additional experimental details for convexity

## E.1    Pipeline

A visual summary of the PH pipeline used for convexity detection (Section 5) is given in Figure 19. Every shape has at least one 0-dimensional cycle, i.e., connected component. For the given example point cloud, PD on the cubical complexes weighted by the tubular filtration from the line passing through the bottom of the image will have a second persistent connected component. A positive persistence of the second most persisting cycle for at least some line indicates concavity.

We note here that the convexity could also be detected with PH with respect to the Vietoris-Rips filtration, with some important adjustments. Indeed, [31, Theorem 2] provides a guarantee PH of any function $f$ and shape $S$ can be estimated using an algebraic construction based on Rips complexes from a point cloud $X$ which is a geodesic dense-enough sample of $S$ (and Theorems 3 and 4 in this paper obtain guarantees in scenarios where both function values and pairwise distances are approximate, i.e., defined on the point cloud). To do so to detect convexity, we cannot employ the standard (so-called vanilla) Vietoris-Rips simplicial complex that relies on the distance function, since all point cloud points show immediately at $b = 0$ in the filtration (that all soon get connected into a single component), so that it never sees the two connected components in concave shapes, at any scale $r \in \mathbb{R}$, which are captured with cubical complexes. Filtering the point cloud points by their height (yielding a so called weighted Rips filtration) might capture the multiple connected components, but these components can get connected with an edge as soon as they are born, if they are close to each other with respect to Euclidean distance (Figure 20). This can be resolved by considering the geodesic distance (the length of the shortest path along the manifold, or a graph), which will allow the multiple connected components to persist longer in the filtration. Indeed, the geodesic distance between points in the "disconnected" regions in concave shapes (the clusters) is larger than the Euclidean distance, so that these only get connected later in the filtration.

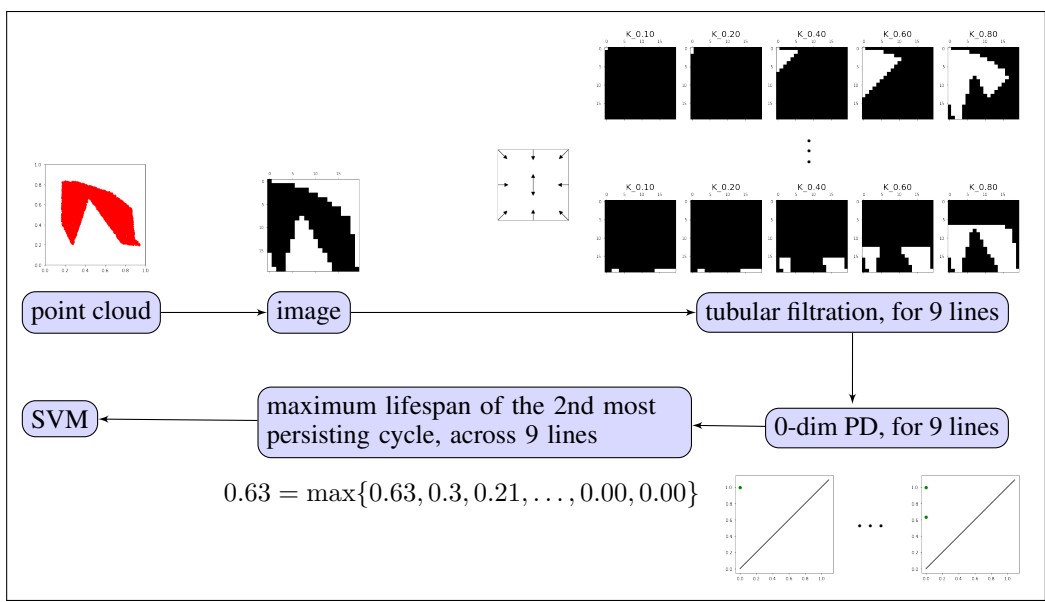

Figure 19: Persistent homology pipeline to detect convexity.

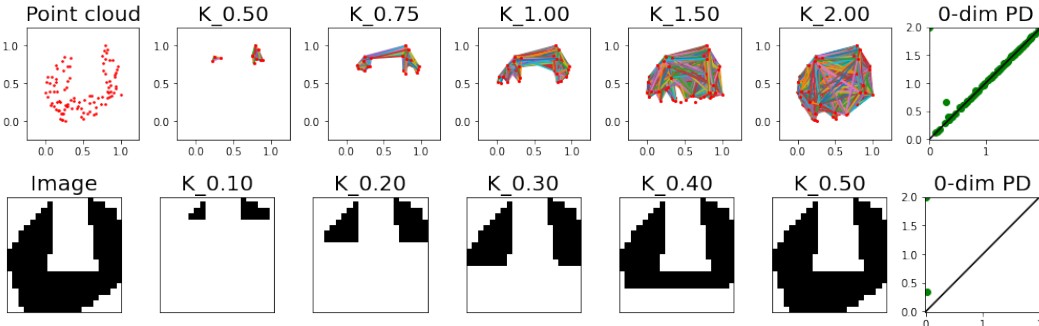

Figure 20: Convexity detection with PH on simplicial and cubical complexes. The concavity can be detected with the weighted Vietoris-Rips simplicial complex, with the tubular filtration function on the vertices (in this figure, with respect to the horizontal line at the top of the point cloud). The filtration function on edges is defined according to the Euclidean distances, but in way that ensures that an edge can only appear in the filtration after both vertices incident to this edge appear in the filtration (for details, see [5]). However, these multiple connected components can still be connected with an edge, if they are close in the Euclidean space. This could be circumvented by considering the weighted Vietoris-Rips which relies on the geodesic distances (which are expensive to compute), or by considering cubical complexes instead, where the connected components remain separate until they merge with the rest of the shape.

The important thing to keep in mind is to choose a filtration that will see disconnected components for concave shapes. We choose cubical complexes as they are more straightforward and do not involve the calculation of geodesic distances. Indeed, as the authors of [31] note, geodesic distances are not known in advance and have to be estimated through some neighborhood graph distance, and computing full pairwise geodesic distances is expensive [64, 74] (e.g., there are deep learning efforts to estimate these geodesic distances on point clouds, such as [54, 86]).

Finally, we also note that the calculation of geodesic distances for curvature detection (Section 4) was straightforward since the point clouds were sampled from unit disks from manifolds with constant curvature, which enabled us to directly rely on the analytical formulas for geodesic distance.

Table 4: Accuracy across multiple runs for convexity detection.

| Experimental setting | Run | PH | ML | NN shallow | NN deep | PointNet |
|---|---|---|---|---|---|---|
| train = regular
test = regular | run 1 | **1.00** | 0.74 | 0.72 | 0.72 | **1.00** |
| | run 2 | **1.00** | 0.74 | 0.77 | 0.60 | **1.00** |
| | run 3 | **1.00** | 0.74 | 0.75 | 0.75 | **1.00** |
| | mean | **1.00** | 0.74 | 0.75 | 0.69 | **1.00** |
| | std dev | 0.00 | 0.00 | 0.02 | 0.08 | 0.00 |
| train = random
test = random | run 1 | **0.85** | 0.56 | 0.60 | 0.46 | 0.40 |
| | run 2 | **0.85** | 0.56 | 0.55 | 0.52 | 0.61 |
| | run 3 | **0.85** | 0.56 | 0.54 | 0.59 | 0.71 |
| | mean | **0.85** | 0.56 | 0.56 | 0.52 | 0.57 |
| | std dev | 0.00 | 0.00 | 0.03 | 0.06 | 0.16 |
| train = regular
test = random | run 1 | **0.78** | 0.59 | 0.59 | 0.59 | 0.51 |
| | run 2 | **0.78** | 0.59 | 0.56 | 0.60 | 0.47 |
| | run 3 | **0.78** | 0.59 | 0.57 | 0.57 | 0.57 |
| | mean | **0.78** | 0.59 | 0.57 | 0.59 | 0.52 |
| | std dev | 0.00 | 0.00 | 0.02 | 0.08 | 0.05 |
| train = random
test = regular | run 1 | **0.96** | 0.54 | 0.55 | 0.49 | 0.54 |
| | run 2 | **0.96** | 0.54 | 0.54 | 0.42 | 0.57 |
| | run 3 | **0.96** | 0.54 | 0.56 | 0.52 | 0.52 |
| | mean | **0.96** | 0.54 | 0.55 | 0.48 | 0.54 |
| | std dev | 0.00 | 0.00 | 0.01 | 0.05 | 0.02 |

## E.2 Performance across multiple runs

For any experimental run, PH is better able to distinguish between convex and concave shapes than the other machine- and deep-learning pipelines (Table 4).

## E.3 Computational resources

Results related to the computational efficiency of the different approaches (trained on regular, and tested on regular shapes) are summarized in Figure 21. In this case, the PH pipeline significantly outperforms the other methods, since it relies on a scalar summary of a point cloud (the maximum lifespan of the second most persisting connected component, across 9 tubular filtration function lines, see Section 5 and Figure 19). On the other hand, PointNet relies on raw point clouds and therefore has a very high memory consumption, since point clouds have 5 000 points for this task (compared to 1 000 points for the detection of the number of holes, or 500 points for curvature detection).

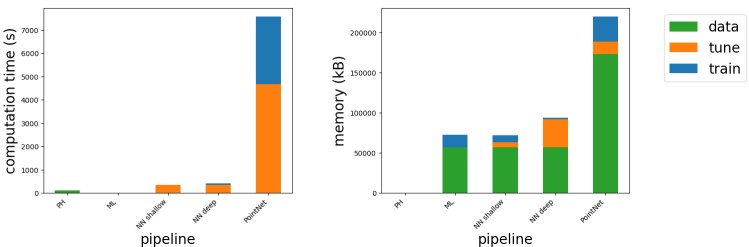

Figure 21: Computational resources for the detection of convexity.

## E.4 Mislabeled point clouds

In order to gain a better understanding of the performance and limitations of our PH pipeline, we look at some examples of mislabeled point clouds. Figure 22 shows a few point clouds sampled from concave shapes that are erroneously classified as convex by PH pipeline (trained on regular, and

tested on random shapes). The figure also clearly suggests that considering additional lines for the tubular filtration function would resolve these issues.

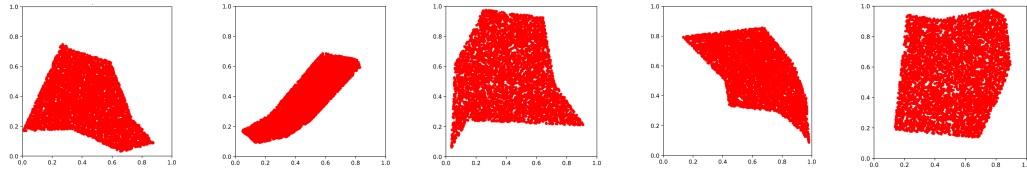

Figure 22: Examples of point clouds from concave shapes incorrectly classified as convex.

# F   Guidelines for persistent homology in applications

Our results demonstrate that PH can be successful in applications for which detecting the number of holes, curvature and convexity is important. Based on our findings, we delineate guidelines for the choice of filtrations and signatures, the input and output of PH pipelines, and draw a better understanding of the topology and geometry properties that are captured by long and short persistence intervals (see Figure 23).

We again note here that we use the alpha simplicial complex for the detection of number of holes in order to improve the computational efficiency, but that the same can be done with the standard Vietoris-Rips filtration. In addition, we discuss in Appendix E.1 that convexity can alternatively be detected with the weighted Vietoris-Rips filtration, filtered by the tubular function, and relying on geodesic distances.

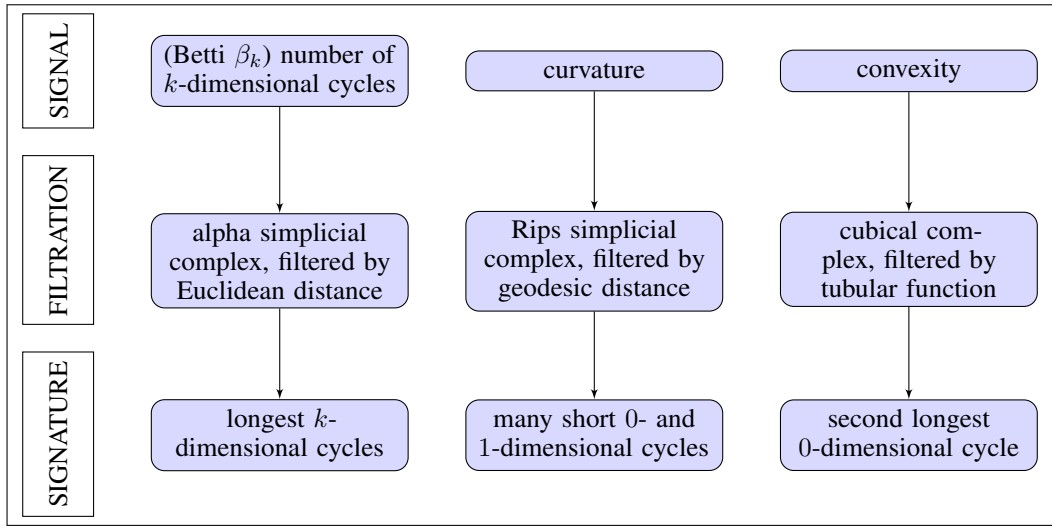

Figure 23: Persistent homology can be useful in applications where $k$-dimensional cycles, curvature or convexity are important features. The choice of filtration and persistence signature, including the focus on the long and/or short persistence intervals, depends on the signal of the particular application.

## F.1   Adjustments of PH pipeline for related applications

Some obvious adjustments to the guidelines from Figure 23 can be made for applications related to the ones that we consider here. Some possible adjustments include the following.

- If it is expected that the data set is noisy, the suggested filtration function should be weighted by density to achieve robustness to noise (as in Section 3).

- In our experiments, we focused our attention on the PH information relevant to the individual problem at hand, but for other related applications, one might need to consider a different type of information given by PH. For example, if we do not only aim to distinguish between convex and

concave shapes, but rather to capture more information about the possibly many concavities, we should not restrict our attention only to the second most persisting cycle, nor consider the maximum across filtration function directions. Instead, it would be useful to take all PH intervals into account for such an application.

- If there are multiple sources of differences in the data, it can be a good idea to combine the different pipelines. For example, if two classes can be differentiated with some concavities, 0-dimensional PH on the tubular filtration will be useful, but if it is also the shape curvature that can help make a distinction, this information can be concatenated with 0- and 1-dimensional PH on Vietoris-Rips filtration.

## F.2 Discussion of PH pipeline for other applications

**Step 1: Signal** Figure 23 and the discussion above clearly indicate that, when faced with a new problem, it is essential to first try to identify the important information, the signal. To illustrate this more clearly, we list some examples of very different types of signal in Appendix F.2.1, Appendix F.2.2, Appendix F.2.3. Once there is some understanding of the signal, the next steps are to choose the filtration and signature accordingly.

**Step 2: Filtration** The aim is for the filtration to capture the signal. For instance, the Vietoris-Rips filtration encodes the size of cycles, while the height or tubular filtration encodes their position. The choice of filtration also influences which type of geometric properties will be captured by long or short persistence intervals. To illustrate the importance of the choice of filtration for the interpretation of long and short intervals, we consider the example point cloud in Figure 10. PH with respect to any meaningful filtration can detect the topology of the underlying shape, i.e., the two holes. However, for the height filtration function from the top of the image, the small circle would have a longer lifespan of the two (as it is born earlier in the filtration), and the large circle can have a seemingly very short lifespan (as it is only born at the bottom of the image). For the Vietoris-Rips filtration it is the opposite (a small cycle has short persistence), and PD on the height filtration from the bottom of the image would see cycles of comparable persistence.

**Step 3: Signature** The choice of persistence signature and the corresponding metric further influences the emphasis on long or short persistence intervals. The Wasserstein distances [20] between PDs place more importance to long persistence, and the same is true for $L_p$ or $l_p$ distances between other common choices of persistence signatures, with the standard choice of parameters. However, similarly to our discussion above in F.1, one might want to focus only on short intervals, e.g., by considering only the intervals with lifespan below a certain threshold (so that the distances would be computed between this simplified PH information). Some alternative ways to give more weight to short intervals, or intervals of any persistence, or even birth value, is via persistence images with the appropriate choice of the weighing function [2], or stable ranks with appropriate densities to prioritize [24]. Note, however, that the stability results also depend on the choice of filtration, persistence signature and metric [103]. Finally, we note that there has recently been a lot of effort in trying to train neural networks to learn what the best PH signature is for specific types of applications [21, 42, 75, 92].

In the remainder of this section, we consider a few hypothetical applications to discuss the relevance of signal, filtration and signature that we hope will be useful for practitioners. In particular, the examples highlight that the importance of long and short persistence intervals depend on the particular application domain. In this context, it is sensible to try to understand the nature of information that is captured with PH (e.g., topological or geometric, any of which might or not be important). The examples thus help us to refine an ongoing discussion in the field about the information detected by intervals of a specific length in a PD:

- **Long persistence intervals as signal.** Indeed, this is true in examples from Figure 24 (when long intervals capture important topology) or Figure 27 (where long intervals capture important geometry). However, an example in Figure 28 (together with results in Section 4 and Section 5) highlights that important information can be encoded in short intervals.

- **Short persistence intervals as noise.** Figure 28 is an example where important information is captured by short intervals (or in this case, a short interval). This can also be seen in the experimental results in Section 4 and Section 5.

- **Long intervals capture topology.** An example in Figure 27 highlights that long intervals, next to topology, also capture geometric information.
- **Many short intervals capture geometry.** An example in Figure 28 (together with results in Section 5) shows that even a single short interval can capture (important) geometry.

### F.2.1 Topology is important, geometry is irrelevant

In some applications, it might be useful to make no distinctions between a circle, a circle with a dent, the circle under translation or scaling, or a square (Figure 24). In this case, "shape" is understood through the lens of topology — more precisely, what we are interested in is what is called "homology-type" —, where one object can be deformed into another by bending, shrinking and expanding, but not tearing or gluing. Indeed, to a topologist, a coffee mug and a donut have the same shape.

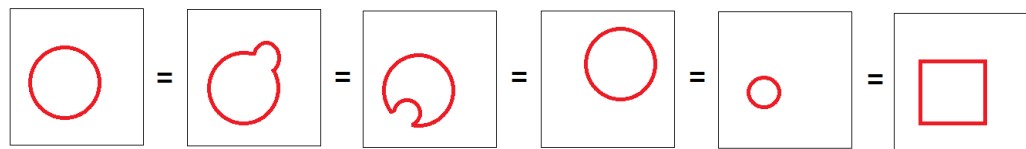

Figure 24: An example of an application where topology is the signal. We consider all the shapes to be the same, i.e., to represent the same class of data, as they all have one connected component and one hole.

It is possible to obtain the same PH summaries for all of the shapes from Figure 24. Indeed, 1-dimensional PDs with respect to the standard Vietoris-Rips filtration on a unit circle and a unit square sampled with same density (reflected in the birth values) could respectively be $\{(0.1, 1)\}$ and $\{(0.1, 1.41)\}$, since the death value reflects the size of the hole. However, we can focus on the cardinality $|\text{PD}|$ of PDs, that here only encodes topological information. Alternatively, we could rather consider PDs calculated on cubical complexes filtered by the binary or grayscale filtration.

Let us further assume that point clouds with multiple holes might be present in the data, but that the only relevant information is the presence of holes, and not their number (Figure 25). An example of such application could be classification between chaotic and periodic (biological) time series, since the circularity of the so-called Taken's embedding point cloud reflect periodicity of the underlying time series [83]. In this case, we can only focus on the maximum persistence $\max\{l = d - b \mid (b, d) \in \text{PD}\}$. Although PD captures a lot of topology and geometry, this choice of summary obviously ignores a lot of this information.

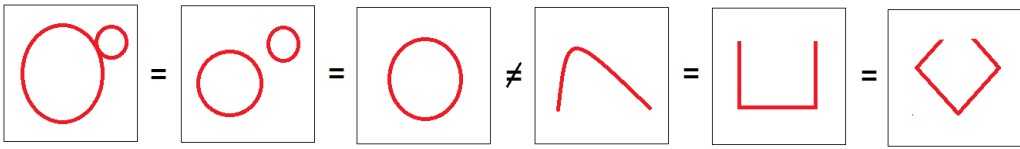

Figure 25: An example of an application where the presence of holes is the signal. The first three shapes in the left part of the figure belong to the same class as there is at least one hole present, whereas the remaining three shapes belong to another class with no holes.

### F.2.2 Topology is irrelevant, geometry is important

For other type of applications, the shapes from Figure 24 might be representatives of different classes of objects (Figure 26). Since they all have a single connected component and a single hole, the topological information has no use in discriminating between the classes. However, the geometric information about their size and position is useful.

**Important geometry encoded in long intervals** Consider an example where every shape in the data set has only two holes (Figure 27), and PH with respect to the Vietoris-Rips filtration. The two longest intervals reflect these two holes (topological information), but their lifespans reflect their size, and it is this geometric signal that can help discriminate between the shapes.

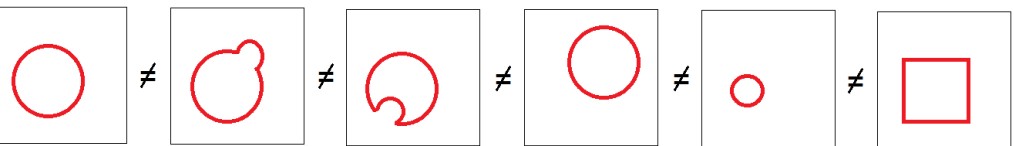

Figure 26: An example of an application where geometry is the signal. In this case, every shape in the figure represents a different object, i.e., they all belong to different data classes.

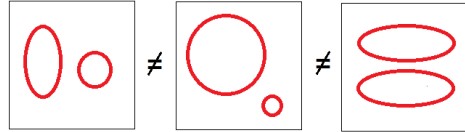

Figure 27: An example of an application where the size of the holes is the signal. The three shapes all have two holes, but their size is meaningful for this application, so that they all belong to different data classes.

**Important geometry encoded in a single interval with the shortest persistence**  We consider a hypothetical cancer-detection application. Let us assume that the data set consists of medical images of some cells in the human body, which look like a certain number of holes (e.g., a grid-like structure). Now imagine that the only difference between the healthy and cancerous cells is the presence of a tiny hole somewhere in the image (which might correspond to some developing cancerous tissue) (Figure 28). For PH with the Vietoris-Rips filtration, the lifespan of each cycle registers its size, but it is the very short persistence of the tiniest holes which would be the most important for this application, as it would be this local geometry signal that would allow to discriminate between the two classes of data, i.e., to detect the presence of cancer.

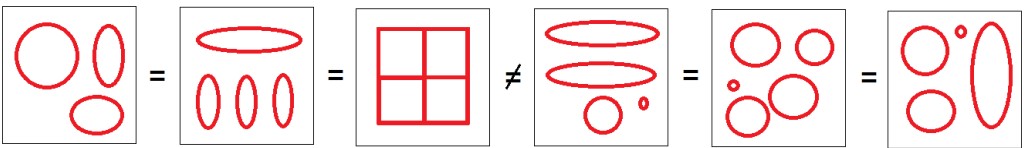

Figure 28: An example of an application where the presence of a tiny hole is the signal. The first three shapes in the left part of the figure reflect the images of healthy cells, whereas the remaining three shapes indicate developing cancerous tissue.

For example, PH for healthy and cancerous cells can respectively have lifespans $(20, 20, 20, 20, 0)$ and $(20, 20, 20, 20, 0.05)$. The stability theorems imply that the difference between the PH on the healthy and cancerous cells is "small" (or more precisely, it is limited by the difference in their filtrations), but this difference is important for this problem and hence not "noise."

PH can be successful for this task even if the number and size of holes varies across images of healthy cells. In this case, the lifespans for PH of healthy cells could, e.g., be $(20, 15, 12, 25, 0)$, $(13, 21, 15, 17, 0)$, and $(14, 15, 27, 20, 0.05)$, $(19, 21, 15, 17, 0.05)$, for cancerous cells. Here, the distance between the PH for healthy and cancerous cells is overwhelmed by the distance between the long cycles, that reflect irrelevant information for the problem. However, we could consider a PH signature that only focuses on short intervals, or choose PIs that give a greater weight to short intervals. Alternatively, if there is a number of labeled images available, the difference with respect to the short persistence interval can be learned.

In the same way, it might be the case that images can only be distinguished with a hole of medium persistence, and therefore the importance of different lifespans depends on the application, i.e., data set. If we know this a priori, we can use PIs and give the greatest weight to the intervals with the most distinctive persistence.

### F.2.3 Topology and geometry are important

To conclude our guidelines, we consider an example of an application in which both geometric and topological information are important. Let us consider a classification problem where the shapes in Figure 29 represent different objects, i.e., different data classes. In this case, it is topology and geometry together that provide useful information. The standard choice of PH on the Vietoris-Rips filtration can help to distinguish between these objects. The PH signature should consider all persistence intervals, since, as discussed in Section 6, geometry (reflecting the size of cycles) is captured in every persistence interval, while the topology is reflected in the number of long-enough intervals.

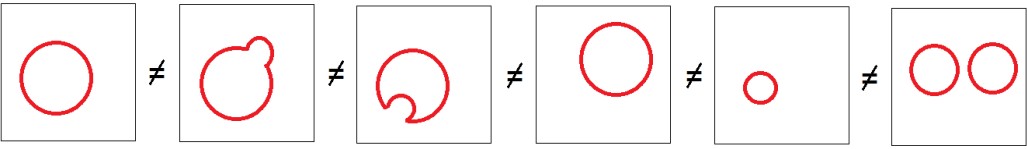

Figure 29: An example of an application where topology and geometry are both signal.

## G  Persistent homology detects convexity in FLAVIA data set

In this section, we employ PH on the FLAVIA data set which consists of $1\,907$ $1200 \times 1600$ images of plant leaves [114]. Figure 30 shows a few examples of images in this data set. The goal of these experiments is to show that PH can be effective on real-world data, but also to illustrate the above guidelines about the appropriate choice of filtration and signature for a given application, and the importance of long and short intervals (Appendix F). We focus on convexity detection, as this is the main contribution of our work.

We classify the leaves according to following measure of convexity which has been shown to be useful for plant species recognition [61]:

$$c(X) = \frac{\text{area}(X)}{\text{area}(CH(X)))},\tag{1}$$

where $CH(X)$ is the convex hull of image $X$. The convexity measure (1) is the most widely used in the literature, and appears in textbooks [124]. Note that we only use the above formula to properly label the data set (Figure 30), but that, deriving convexity information in such a way involves employing a convex hull algorithm.

In the simpler scenario of a binary classification between convex or concave shapes (i.e., signal is the simple: convex - yes or no), we could rely on the same pipeline as in Section 5, where we consider the lifespan of the second most persisting connected component, and then store the maximum such value across all 9 tubular filtration lines (Figure 19, Figure 31). This is sufficient information, since we are only interested in whether PH sees multiple connected components - source of a concavity, for *at least one* line. However, the convexity measure (1), the signal in our application, provides a more detailed level of information (regression problem), so that we keep the lifespan of the second most persisting connected component *for all* lines, in order to capture information about sources of concavities seen with respect to any of the lines.

Moreover, since the convexity measure is calculated relative to the size of the leaf, the tubular filtration *directions* remain the same, but 8 filtration lines pass through the corners of the leaf rather than the corners of the image (Figure 19), and the lifespans are normalized relative to the area of the leaf (total number of black pixels in the binary image). In this way, PH depicts information about concavities for any line and relative to the leaf size, and it is invariant under translation and scaling. We use $30 \times 30$ cubical complexes (on binary images), to capture a higher level of detail for the leaves of different convexity, in comparison with the $20 \times 20$ resolution for the cruder differences in our synthetic data set in Section 5. Figure 31 visualizes the tubular filtration for the 9 different lines, and the resulting 0-dimensional PDs, for an example leaf image (bottom image from Figure 30).

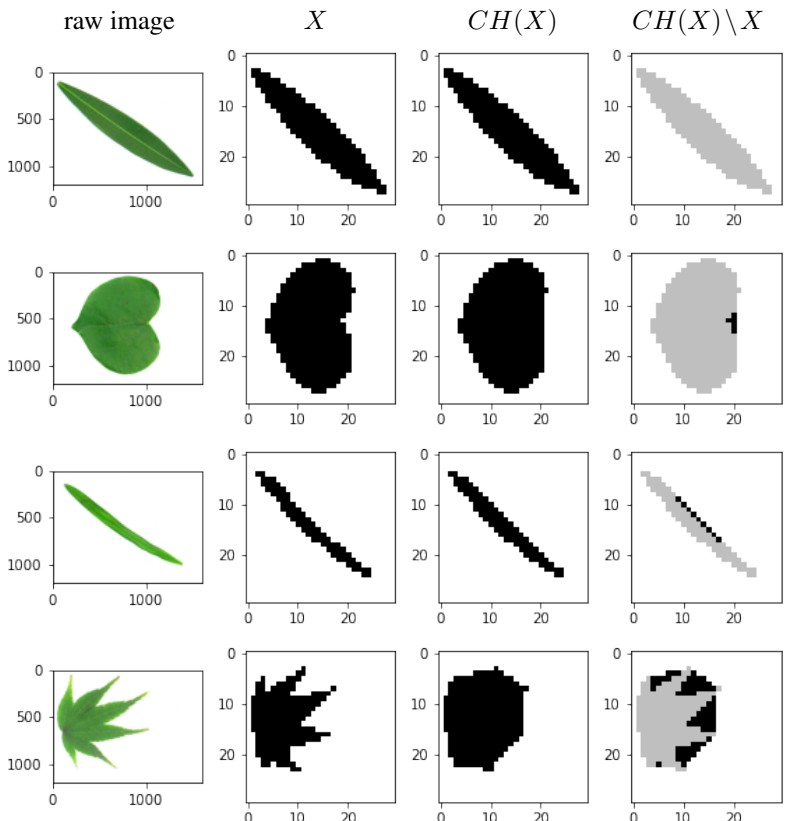

Figure 30: A few example images from the FLAVIA leaf data set. The images are shown, from top to bottom, with a decreasing label, i.e., convexity measure $c(X)$, 1.00, 0.98, 0.89, 0.71. Note that the second image from the top is more convex than the third image, since the considered convexity measure $c(X)$ is given relative to the area of the leaf. The PH lifespans on the tubular filtration with respect to 9 different lines are respectively from top to bottom, [0.00, 0.00, 0.00, 0.00, 0.00, 0.00, 0.00, 0.00, 0.00], [0.00, 0.00, 0.00, 0.00, 0.00, 0.00, 0.00, 0.00, 0.00], [0.00, 0.00, 1.09, 0.00, 0.00, 0.00, 0.00, 0.00, 0.00], [0.00, 2.52, 10.08, 3.78, 0.00, 1.26, 0.00, 0.00, 0.00].

Linear regression on the FLAVIA data set, trained on 70% of random images, with each image represented with the 9-dimensional vector of lifespans of the second most persisting component across all tubular filtration lines, obtains a mean square error of 0.00065. The regression line in Figure 32 shows that PH is effective in classifying the FLAVIA leaves according to a measure of convexity. The convexity of some thin leaves (such as the image in the third panel in Figure 30) gets overestimated with our PH pipeline, since concavity is not captured well with our crude resolution, that could easily be improved.

Furthermore, even more detailed information can be captured if the lifespans of the third, fourth, ... most persistent connected component would be kept, because some leaves have more than two sources of concavity for a single line, that result in more than two connected components. For example, the 0-dimensional PD of the image in Figure 31 has more than two persistence intervals for some tubular filtration lines. The accuracy can thus be improved by considering the lifespans of *all* short intervals (and across all lines), and again, by considering more tubular filtration lines (Section 5). This clearly illustrates how the choice of filtration and signature, the input and output of PH, should be guided by the signal in the given application. Moreover, it shows that one short interval can be sufficient for some applications, but that in other cases, many short intervals might store the needed additional level of (geometric) information.

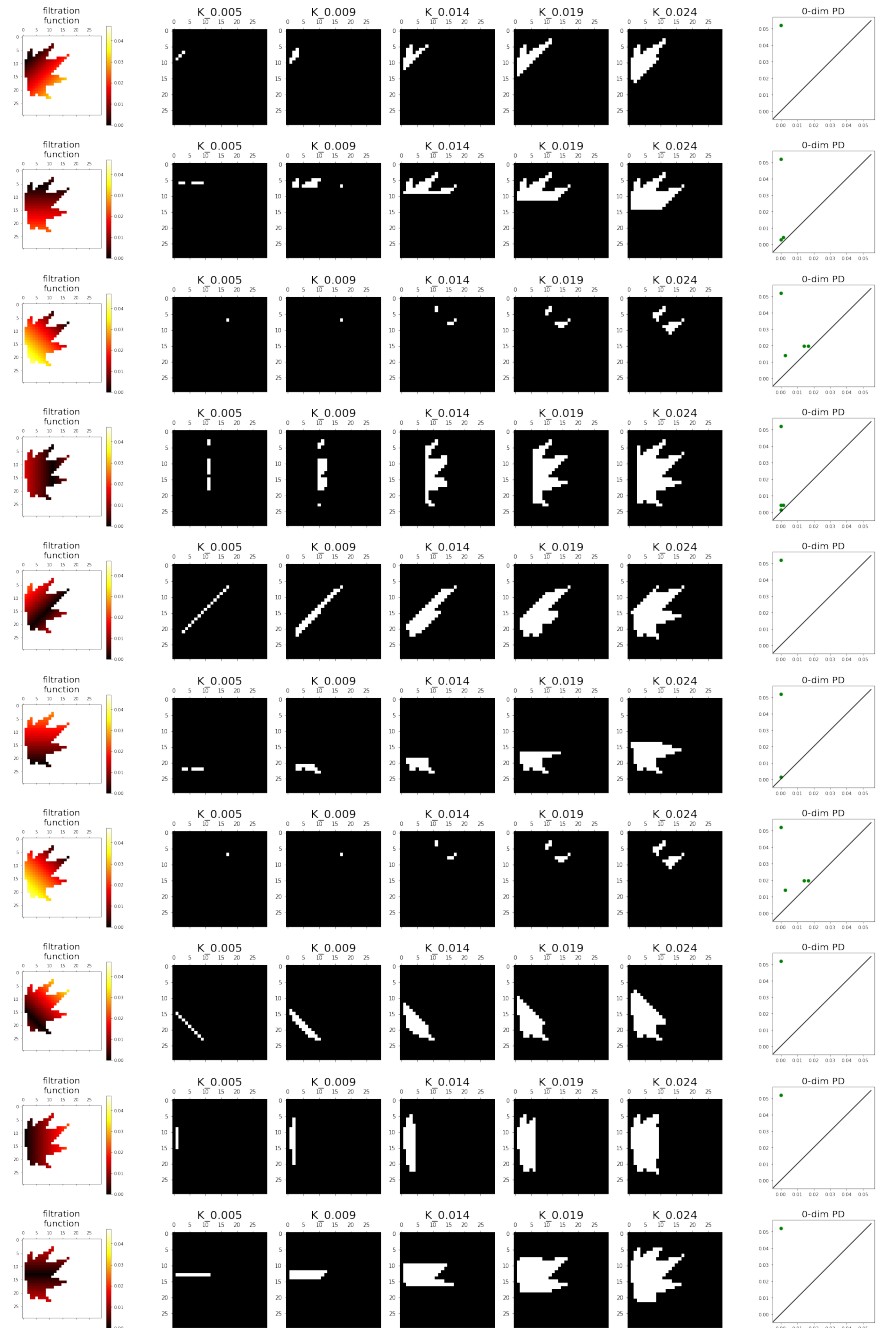

Figure 31: The tubular filtration for the 9 considered lines, and the resulting 0-dimensional PDs for an example image. The concavity is detected with multiple connected components that are seen for a few lines.

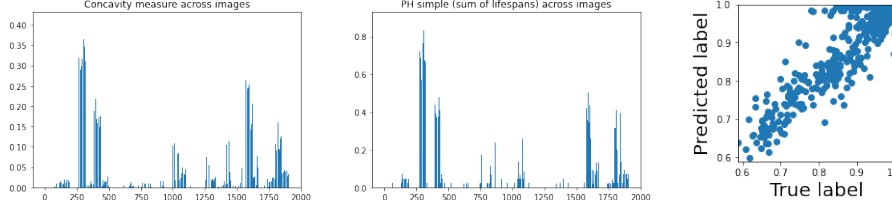

Figure 32: Results on the FLAVIA data set. The first two plots show that there is a good correspondence between the concavity measure $(1 - c(X))$ (left panel) and the simple PH signature that only considers the sum of lifespans across the tubular filtration lines (middle panel). The regression line on lifespans from all 9 tubular filtration lines shows good performance of our PH pipeline.