# OpenReview forum: "On the Effectiveness of Persistent Homology"
_NeurIPS.cc/2022/Conference — NeurIPS 2022 Accept_

### Official Review · Reviewer_6onG · 2022-07-06

**Rating:** 6
**Confidence:** 4
**Soundness:** 3 good
**Presentation:** 3 good
**Contribution:** 3 good

**Summary:**

The paper mainly discusses what can be detected by persistent homology (PH) features by introducing three shape analysis tasks. Experiments on these tasks show that PH can effectively detect these shapes even with limited computational resources and limited training data.

**Questions:**

See "Weakness".

**Limitations:**

The limitations are adequately addressed.

**Strengths And Weaknesses:**

In general, the paper does add some interesting and new discussions, but it seems that some discussions have been covered in previous works.

Strengths:

The paper is generally well-written, and not hard to follow, except for some minor mistakes. E.g., “0- and 1-dim PH” in Figure 1 should be replaced by “0- and 1-dim PI”; In Line 282, it seems that the longest interval should be topological noise, while the second-longest one is useful.

The discussion in the paper is interesting. Recently, there are many works integrating PH into various downstream deep learning tasks and achieving state-of-the-art results. However, many works lack insight on why PH can boom performance on these tasks. Therefore, researching what can be detected by PH quite interests me. The discussion on long/short persistence is also interesting.

The discussion on using PH to detect convexity is new.

Weakness:

First, detecting the number of holes with PH is not new. Actually, we can directly compute the betti number (the number of holes) of a given point cloud data. (E.g., in [56], researchers compute the betti number with several preprocessing steps.) Then what is the superiority of the proposed method: (to infer/learn the number of holes with PH features)?

Second, detecting curvature is not new either. Actually, a similar experiment that uses PH to detect curvature is done in [17]. In fact, they also provide theoretical guarantees and experiments on unsupervised learning.

Also, there are some comparably minor problems. (1) building simplicial complexes and computing PH is quite computational costly, therefore better add comparison on times in the experiment part; (2) adding persistent homology on downstream tasks that require topological information such as curvature/convexity/betti number can better support the experiment part of the paper. But considering the limited rebuttal time, this can be future work.

--**Post rebuttal**--

Generally speaking, the topic of the paper is worth researching, and some of the discussions are interesting. Although most of the experiments are based on synthetic datasets, it should be admitted that very few real-world graphs totally rely on the number of holes/convexity/curvature to predict certain attributes, e.g., their label.  Therefore I would like to raise my score, but certain improvements should be considered:

(1) Add PH to deep learning models and evaluate whether it can boost performance on real-world tasks. For example, over-squashing is a curvature-related attribute [1], therefore it can be evaluated whether adding PH in GNN can solve the problem.

(2) Some works combining PH with deep learning models are missing, e.g., [2], [3].

[1] Topping J, Di Giovanni F, Chamberlain B P, et al. Understanding over-squashing and bottlenecks on graphs via curvature. ICLR 2021.

[2] Hu X, Li F, Samaras D, et al. Topology-preserving deep image segmentation. Neurips 2019.

[3] Yan Z, Ma T, Gao L, et al. Link prediction with persistent homology: An interactive view. ICML 2021.

---

> ### Author Response · Authors · 2022-08-02
> **Detection of Betti numbers with PH**
>
> Thank you for pointing out the relevance of our work, and for noting down the weaknesses in detail. Addressing these weaknesses has helped us to make the paper stronger, and its contributions much clearer.
>
>
> Firstly, we do not suggest anywhere in the paper that detecting the number of holes with PH is new. In the Related works (p2 lines 50-52 in the revised manuscript), we had clearly stated that there is theoretical evidence that PH can detect the number of holes (and now we make these results more explicit in Theorem 6). We had then also listed some papers that demonstrate this for some interesting examples, but, that we are not familiar with any work that includes extensive experiments to show that PH can be used for this task (p2 lines 53-57 in the revised manuscript).
>
> The article [56] ([68] in the revised manuscript) is an important one, but that does not **focus** on detection of the number of holes with PH. The goal of this paper is to show that neural networks (with particular choices of activation function) simplify the topology of data, sharply reducing the Betti numbers to trivial values. To do so, the authors do use PH to estimate the Betti numbers of the data as it passes thorough the layers of the neural network, but they never investigate the behavior or precision of this estimation. The several preprocessing steps that you mention are here crucial. As the authors state, ``actual computation of homology from a point cloud data set in practice is more involved than what one might surmise from the description in the last few sections", and they preprocess the data in the following way: (i) they smooth out $X$ and discard outliers to reduce noise, (ii) they **select** the filtration threshold or scale $r \in \mathbb{R}$ and construct the corresponding Vietoris–Rips complex $VR(X, r).$ As the authors note, **this can change the homology (Betti numbers)**. For simulated data, where they know the topology of the underlying manifold completely as they generated it as part of their data sets, the authors employ an iterative heuristic to pick the right scale $r$ (and also, simultaneously, a distance parameter $k$) to compute the homology at this scale (Section 6.3). This heuristic complicates the procedure of obtaining the Betti numbers, and actually, the benefits of persistent homology (over homology) - that it captures homology across scales, is not even exploited here. For the real data (Section 8), however, the topology of the data is not already known, so that the authors do determine this with persistent homology in this scenario. However, the focus in this paper (unlike ours) is not to evaluate the correctness of this step, rather, this information is used as some kind of ground truth (to evaluate the behavior of NN). In our paper, (where we avoid the preprocessing steps above, by incorporating the noise reduction within the PH pipeline with DTM filtration, any by considering PH - homology across scales), we precisely focus on the effectiveness of PH to perform this step well. Thank you for raising this question, we now also list this article in the Related works (p2 lines 57-59) and comment how it differs from our work, making the goal of our paper thereby clearer.
>
> We also do not claim the superiority of PH compared to other methods in the literature (in particular, to the state-of-the-art for each of the problems, as noted in the limitations, p9 line 313-315) - our goal was rather to gain some knowledge into the type of problems that can be solved with PH, as this is not yet well understood. We should point out that we do not necessarily expect that on well-specified mathematical tasks PH will beat state-of-the-art algorithms that have been specifically designed for those tasks, particularly detecting convexity, since there is ultimately no free lunch. Instead, what we think is interesting and remarkable is that PH can in fact solve such tasks it is not specifically or uniquely designed for. To make this point, we compared different implementations of PH against three baselines, SVMs, MLPs, PointNet, with a multitude of hyperparameters for the baselines and diverse conditions on the data. The reported absolute performance and comparisons make it sufficiently clear that PH can solve these tasks, which is what we claimed. Although we agree that a detailed comparison with state-of-the-art methods is an interesting direction to investigate, our contribution is to identify basic tasks that PH can solve and which can better inform us about why and when PH can be effective in practical tasks.
>
> [56] Gregory Naitzat, Andrey Zhitnikov, and Lek-Heng Lim. Topology of deep neural networks. J. 551 Mach. Learn. Res., 21(184):1–40, 2020.

---

> ### Author Response · Authors · 2022-08-02
> **Curvature detection with PH, minor problems and mistakes**
>
> Secondly, you are completely right that curvature detection with PH is not new: We had provided the reference [17] (and also, [83]) on p2 lines 60-64, and on p5 lines 190-193, we again make it clear that we replicate a similar experiment as in the original paper (with some additional experiments to gain insights into the importance of many short intervals). In the revised manuscript, we also make it more explicit that [17] also includes theoretical guarantees, please see Theorem 7.
>
> [17] Peter Bubenik, Michael Hull, Dhruv Patel, and Benjamin Whittle. Persistent homology detects
> 441 curvature. Inverse Problems, 36(2):025008, 2020.
>
> [83] Oliver Vipond. Multiparameter persistence landscapes. J. Mach. Learn. Res., 21(61):1–38,
> 647 2020
>
>
> Next, we adress the minor problems.
>
> > Building simplicial complexes and computing PH is quite computational costly, therefore better add comparison on times in the experiment part.
>
> We had included a comparison on both computation time and memory for each of the problems (Figures 14, 17 and 20 in the revised manuscript), precisely because, as you correctly point out, PH can often be computationally costly. Due to a page number limit, these figures are included and interpreted in detail in appendices, but we do briefly comment on them in the main text at the end of each of the problem sections (p5 line 179-181, p6 lines 218-220, p8 line 282). Moreover, we now discuss theoretical complexity of PH in the new Appendix A.4.
>
> > Adding persistent homology on downstream tasks that require topological information such as curvature/convexity/betti number can better support the experiment part of the paper. But considering the limited rebuttal time, this can be future work.
>
> We completely agree that adding PH to downstream tasks that require topological information such as curvature/convexity/Betti number can better support the paper, and we therefore now include additional experiments on a real-world dataset (Appendix G in the revised manuscript). The results on the FLAVIA image dataset of plant leaves show that PH can effectively detect the convexity measure of leaves, please see our response to Reviewer 786m for details.
>
>
> Finally, we comment on two listed minor mistakes.
>
> > “0- and 1-dim PH” in Figure 1 should be replaced by “0- and 1-dim PI”.
>
> The aim of Figure 1 (now moved to Appendix B, please see Figure 9) was to visualize the **general** PH pipeline, and we had therefore intentionally used ``PH", since we use this notation for any persistence signature that can be used in machine learning frameworks (e.g., PIs, PLs, or lifespans in our experiments). This figure shows PIs as an example for the choice of PH, and we now make this more clear in the figure caption.
>
> >  In Line 282, it seems that the longest interval should be topological noise, while the second-longest one is useful.
>
> The second-longest interval is **topological** noise (since all shapes have a single connected component, so that this interval is short), but it captures useful **geometric** information (about convexity). Thank you for pointing out this important potential point of confusion, we improve our explanation in the revised manuscript (p9 lines 313-315 in the revised manuscript).
>
>
> We believe that the addition of theoretical results (Appendix A), experiments on real-world data (Appendix G), and the additional discussions that we included in the paper related to the suggestions you raised here, have significantly strengthened our paper. We hope that this might motivate you to consider raising your assessment of our manuscript, and we will be attentive to any further comments. Thank you very much once again for the constructive feedback.

---

> > ### Comment · Reviewer_6onG · 2022-08-08
> > **Additional questions**
> >
> > Thanks for your response and clarification, I think I can understand the paper better now.
> >
> > However, I am still a little confused about whether the second-longest interval should be topological noise or not. As you mention, all shapes have a single connected component. Then the persistence point of the whole connected component should contain either (1) (for ordinary persistent homology) the smallest birth time and an infinity death time; or (2) (for extended persistent homology) the smallest birth time and the biggest death time. Under all these situations, the persistence point that encodes the whole connected component should have the longest persistence (interval). Then it should be the topological noise (rather than the persistence point with the second-longest interval).

---

> > > ### Author Response · Authors · 2022-08-09
> > > **Additional comment about topological signal and noise**
> > >
> > > > However, I am still a little confused about whether the second-longest interval should be topological noise or not. As you mention, all shapes have a single connected component. Then the persistence point of the whole connected component should contain either (1) (for ordinary persistent homology) the smallest birth time and an infinity death time; or (2) (for extended persistent homology) the smallest birth time and the biggest death time. Under all these situations, the persistence point that encodes the whole connected component should have the longest persistence (interval). Then it should be the topological noise (rather than the persistence point with the second-longest interval).
> > >
> > >
> > > As we write in our manuscript, stability theorems ensure that the longest intervals (with persistence above some threshold) capture information about the topology of the shape. In the case of our convexity detection dataset (Section 5), this will correspond to the single 0-dimensional interval with the longest persistence, since all shapes have one connected component. This longest interval is the topological signal that encodes the whole connected component. (Here, we mean "topological signal" in a general sense, i.e., capturing topology, it does not refer to the signal in a particular application).
> > >
> > > This means, in a precise sense, that all the remaining intervals are topological noise, as they do not encode information about the global topology of the shape. These intervals do, however, as we have come to understand in the last decade, still capture important geometric information/signal, as was the case for the second-longest interval that enabled us to reveal convexity.
> > >
> > > We thank you again for the time you took to read our paper in detail, and for the constructive feedback. Please do let us know if this issue, or any new questions that you might have in the meantime, remain unclear.

---

### Official Review · Reviewer_sPjT · 2022-07-10

**Rating:** 5
**Confidence:** 4
**Soundness:** 2 fair
**Presentation:** 2 fair
**Contribution:** 2 fair

**Summary:**

This paper studies the effectiveness of PH for detection of number of holes, curvature and convexity using point clouds samples from shapes. The analyses are performed on various simple datasets in comparison with baselines providing promising initial results.

**Questions:**

Training curves show that NNs overfit data easily. Could you please repeat these experiments using networks endowed with normalization and regularization methods such as ResNets?

**Limitations:**

Limitations were briefly addressed.

**Strengths And Weaknesses:**

PH is an important method used for topological data analysis. The paper introduces a nice analysis of the employment of PH for several tasks on point clouds. The paper is well written, and analyses are clearly presented. In several benchmark, PH outperforms baseline with promising results for the future work.

The main weakness of the paper is the datasets. The paper can have more impact if the analyses are extended to point clouds of more complicated shapes, especially shapes of real-world objects. Although the current results are interesting, their generalization to more complicated datasets is not clear.  As mentioned in the limitation, a more detailed comparison with state-of-the-art would improve the paper.

---

> ### Author Response · Authors · 2022-08-02
> **Theoretical results and new experiments on real-world data are included**
>
> Thank you so much for noting the promising aspects of our work, and more importantly, for raising crucial weaknesses that we discuss below.
>
>
> Firstly, we have now run additional experiments on a real-world dataset (Appendix G in the revised manuscript). The results on the FLAVIA image dataset of plant leaves show that PH can effectively detect the convexity measure of leaves. Please see our response to Reviewer 768m for more details.
>
>
> Secondly, we now also now make the theoretical guarantees about why PH works for all three classes of problems (Appendix A in the revised manuscript, which also includes a new result - Theorem 1, which shows that PH can detect convexity), that ensure the generalization of PH to new datasets.
>
>
> Moreover, as you correctly noted, the lack of a more detailed comparison with the state-of-the-art had also been explicitly acknowledged as a limitation of our work (p9 313-315). However, as we note there, our goal was not to show the **advantage** of PH to existing models, but rather to demonstrate that PH **can** solve (i.e., is effective for) these problems in practice (as this has not been done in the literature, besides the curvature detection). We should point out that we do not necessarily expect that on well-specified mathematical tasks PH will beat state-of-the-art algorithms that have been specifically designed for those tasks, particularly detecting convexity, since there is ultimately no free lunch. Instead, what we think is interesting and remarkable is that PH can in fact solve such tasks it is not specifically or uniquely designed for. To make this point, we compared different implementations of PH against three baselines, SVMs, MLPs, PointNet, with a multitude of hyperparameters for the baselines and diverse conditions on the data. The reported absolute performance and comparisons make it sufficiently clear that PH can solve these tasks, which is what we claimed. Besides, the theoretical results that we now include (Appendix A) provide a theoretical guarantee of the effectiveness of PH for these problems. Although we agree that a detailed comparison with state-of-the-art methods is an interesting direction to investigate, our contribution is to identify basic tasks that PH can solve and which can better inform us about why and when PH can be effective in practical tasks.
>
>
> Finally, in the next revision, we hope to repeat the experiments for deep learning models, endowing them with normalization and ResNet regularization. Due to the limited rebuttal time, we put priority on addressing the two major limitations mentioned above first.
>
>
> We belive that the added theoretical results (Appendix A), and new experiments on real-world data (Appendix G) have significantly strengthened our work, and hope they might motivate you to reconsider your assessment and raise your rating of our manuscript. We remain attentive to any questions that you would like to discuss, or suggestions for further improvement, and thank you once again for the constructive feedback.

---

### Official Review · Reviewer_pzbB · 2022-07-11

**Rating:** 3
**Confidence:** 5
**Soundness:** 2 fair
**Presentation:** 3 good
**Contribution:** 1 poor

**Summary:**

In this article, the authors propose to investigate the data science tasks where persistent homology performs actually better than standard methods. They focus on three different problems: holes, curvature and convexity detection, and they show that, for each of them, persistent homology leads to better accuracies than neural nets.

**Questions:**

---What are the variances of the accuracies in the figures? Since some experiments involve adding random noise, and since training and test sets are drawn randomly, accuracies should always be computed on several runs, and their means and variances across the runs should be reported.

---Why is PH suited for curvature detection? One can easily get the intuition why PH is good for hole and convexity, but it is less clear concerning curvature. In particular, why is 0-dimensional PH efficient, but not 1-dimensional PH? Why do the short bars matter more than the long ones? It would be nice if the authors provided some more intuition for this task.

---The paper should be more explicit about the parameters: in the hole detection experiment, what are the DTM parameters? In the convexity detection experiment, what is the resolution of the image computed from the point cloud?

---Instead of computing an image from the point cloud in the convexity experiment, could one use scalar field analysis instead (https://link.springer.com/article/10.1007/s00454-011-9360-x)?

---What is the influence of the point cloud dimension? Is there any curse of dimensionality? In other words, is there any reason to believe that the proposed topological methods would still be efficient?

**Limitations:**

I think the authors properly identified the limitations of their experiments.

**Strengths And Weaknesses:**

I think the question of identifying the tasks where computational topology improves on existing methods is definitely important, making the paper well motivated. Moreover,  the writing of the paper is very good and clear.

However, the significance of the paper is limited by two major weaknesses.

First, there is no theoretical statements or results about the topological methods used to solve the tasks. This makes the claims of the paper harder to believe, since there is no way of estimating the generalization capability of topological methods on data sets that are different from those used in the paper.

Second, the experiments of the paper are only done on synthetic data sets in two dimensions. When there is no theory, it is necessary to benchmark many different data sets, from both synthetic and real data, in order for the claims to be convincing, but the proposed experiments are much too synthetic and simple.

---

> ### Author Response · Authors · 2022-08-02
> **Theoretical results and new experiments on real-world data are included**
>
> We appreciate the time you took to read the paper in detail, and to raise important items that we address below. Your comments have helped us to significantly improve our work.
>
>
> Firstly, thanks for rising the point about theory. This is an important aspect, which is why we had already made brief references to some theory behind why PH works for the detection of number of holes and curvature, and some theoretical work somewhat related to our PH pipeline used for convexity detection (p2 lines 48-59, p2 lines 54-55, p8 line 263 footnote in the original manuscript). In the revised manuscript (Appendix A), we have made the theoretical guarantees more explicit, and we also prove a theorem that states that the PH can detect convexity (new result):
>
> **Theorem 1.** Let $X \subset \mathbb{R}^2$ be triangulable. Let $l$ be any line in $\mathbb{R}^2$, and denote by $h_l$ the height function with respect to $l$. For any $r \in \mathbb{R}$, denote by $X_{h_l,r}$ the subset of points at distance at most $r$ from $l$. Then  $X$ is convex if and only if the persistence diagram in degree $0$ of the filtered space $\{X_{h_l,r}\}$ contains exactly one interval.
>
> The idea of the proof is to, for any given concave shape $X \subset \mathbb{R}^2$, identify the height filtration direction for which PH sees multiple connected components (persistence intervals), and thereby detects the concavity. These theorems then also give further support that PH can generalize to new datasets.
>
> Secondly, as also pointed out in our response to Reviewer 786m, we do agree that considering real-world data is an important aspect towards understanding the effectiveness of PH in practice. As a step in this direction, we run experiments on the FLAVIA image dataset of plant leaves (Appendix G in the revised manuscript). The results show that PH is effective in classifying the leaves according to a measure of convexity. We are also adding code for these experiments and in particular a Jupyter notebook for this plant morphology application.
>
> Finally, we address your questions in the next responses.

---

> > ### Comment · Reviewer_pzbB · 2022-08-07
> > **Response to authors**
> >
> > Thank you for your comments. I understand the article better now, and I appreciate the new theoretical result Theorem 1 (minor comment: the height function/filtration wrt a line usually refers to the function obtained after projecting the points onto the line, not the distance to the line, so you might want to change a bit the wording there).
> >
> > However, I will stand by my score, unless my biggest concerns are alleviated. I think it is necessary to provide more experiments on real-world data for each of the problems solved by PH in order to be convincing. Moreover, Theorem 1 is nice but only works for shapes in the Euclidean plane, so I am not sure how much it can be used to show generalization capabilities of PH.

---

> > > ### Author Response · Authors · 2022-08-09
> > > **Additional comments about applications, Part I**
> > >
> > > > I think it is necessary to provide more experiments on real-world data for each of the problems solved by PH in order to be convincing.
> > >
> > > We understand your concern about the lack of more experiments on real-world data, but we must stress that our goal for this paper was not to add to the important literature on the use of PH in such applications (in our Introduction, we cite some of these papers and point to a database that lists 395 applications of PH).
> > >
> > > Rather, as we are also explaining in our response to a similar remark by Reviewer 786m, the title of our manuscript is inspired by a famous paper from 1960, "The unreasonable effectiveness of mathematics in the natural sciences", where Wigner discusses, with wonder, how mathematical concepts have applicability far beyond the context in which they were originally developed. In particular, he argues that there is a need to understand **why** such concepts are useful. The same, we believe, is true for persistent homology. While this method has been applied successfully to a wide range of application problems, we believe that for PH to remain relevant, there is a need to better understand why it is so successful. Thus, we distinguish between the **usefulness** of PH for applications, which has been  attested in hundreds of applications and publications, and its **effectiveness**, namely that PH is capable of producing an intended or desired result.
> > >
> > > A related observation was made by Reviewer 6onG, who noted that "there are recently many works integrating PH into various downstream deep learning tasks and achieving state-of-the-art results, but that many works lack insight on why PH can boom performance on these tasks, making the research on what can be detected by PH relevant'', which is precisely what we aim to do in this work. We have improved the description of our scope better in the Introduction (p1 lines 24-28 of the revised manuscript).
> > >
> > > In our response to a similar remark by Reviewer 786m, we have also illustrated the above discussion with an example application of PH [1], pointed out by the Reviewer. This is an interesting application of PH in molecular biology which provides insights for researchers in the field who want to use PH on similar data (also mentioned by the authors, "the proposed methods can be easily extended to other applications in the structural prediction of biomolecular properties"). In the abstract the authors state that PH reveals the **hidden** structure relationship in biomolecules and in the conclusions that PH characterizes 3D biomolecular structures. This is further detailed in the paper, where it is stated that, intuitively, 0-dimensional PH describes pairwise atomic interactions, and 1- and 2-dimensional PH characterize the hydrophobic network and geometric rings and voids. However, it remains elusive what type of structure was important in this application and only limited insights are given on whether PH would work for different applications. This is further complicated by several application-specific aspects; e.g., persistence is associated with the bond length, ring or cavity size, flexibility and steric effect; and combination of PH with CNNs. It is already known from the definition of PH that it reveals $k$-dimensional cycles (connected components, holes, voids, ...). What we are coming to understand in the last years is the important geometry information captured by PH, such as curvature (first by Bubenik in 2020) and convexity, as we demonstrate in our work, both theoretically and experimentally. In other words, the demonstration of usefulness of PH in real-world applications does not help us to understand if PH, next to the number of holes in specific dimensions, also captured some additional geometric phenomena (for instance, such as the position or size of the holes) that were actually crucial for its good performance.
> > >
> > > [1] Z. X. Cang and Guo-Wei Wei, TopologyNet: Topology based deep convolutional and multi-task neural networks for biomolecular property predictions, PLOS Computational Biology, 13(7), e1005690 (2017).

---

> > > ### Author Response · Authors · 2022-08-09
> > > **Additional comments about applications, Part II**
> > >
> > > It is precisely for the reasons listed in the Part I of our response above that we decided to study three fundamental shape analysis tasks where we have control over the source of difference between classes of data (signal). It was crucial for us consider the data which precisely differs with respect to the topological or geometric notion of interest (Betti number, curvature, convexity). Moreover, the theoretical results that are now explicitly formulated in Appendix A of the revised manuscript (Theorem 1, Theorem 6, Theorem 7) also speak to the generalization of these results to data sets which are different from those used in the paper.
> > >
> > > Having established that PH is effective for detecting holes, curvature and convexity, in the revised manuscript (Appendix G) we now show that PH is effective in detecting convexity in a real-world data set of leaves. Again, here, our intent was not to show that PH can classify the leaves, since this would have left us with the question as to **why** PH can solve this task. We thus carefully selected a measure of convexity that has been shown to be a good classifier for leaves, and we have shown that we can detect the different values of this measure of convexity for leaves using PH.

---

> > > ### Author Response · Authors · 2022-08-09
> > > **Additional comments about Theorem 1**
> > >
> > > > Moreover, Theorem 1 is nice but only works for shapes in the Euclidean plane, so I am not sure how much it can be used to show generalization capabilities of PH.
> > >
> > > Thank you for raising this question. We have indeed put some thought into the generalization of our Theorem 1. Due to the limited rebuttal time, we only included a result for $\mathbb{R}^2$ that corresponds to our experiments (Section 5). However, we believe that Theorem 1 can be generalized to $\mathbb{R}^3$ in the following way.
> > >
> > > **Theorem 1.** Let $X \subset \mathbb{R}^d$ be triangulable, with $d=2$ or $d=3$. Let $l \subseteq \mathbb{R}^{d-1}$ be any hyperplane, and denote by $h_l$ the height function with respect to $l$. For any $r \in \mathbb{R}$, denote by $X_{h_l,r}$ the subset of points at distance at most $r$ from $l$. Then $X$ is convex if and only if the persistence diagram in degree $0$ of the filtered space $\{X_{h_l,r}\}$ contains exactly one interval, and the persistence diagram in degree $1$ is empty.
> > >
> > > The main difference in the proof is that, for concave shapes, we might not always have multiple connected components (detected with PH in homological degree 0). For $q \in \mathbb R^d \setminus X$ on the straight-line segment between points $p_1, p_2 \in X,$ we again consider a ball $B(q, \epsilon)$ placed within this source of concavity (please see Figure 8 in the revised manuscript). However, in this case, the sublevel set $X_{h_l,r}$ might still be path-connected, i.e., there might exist a path between points in $p_1, p_2 \in X$ which lies completely in $X_{h_l,r}$. This path then goes "around the ball" $B(q, \epsilon)$, yielding a loop detectable by PH in homological degree 1.
> > >
> > > In the revised manuscript we will then add Theorem 2 that tackles the more general case. We will also change our wording in the definition of the height filtration function (from lines to hyperplanes), to be applicable to the more general scenario. We are currently investigating what happens for $d \geq 4,$ and we will make sure to explicitly address this in the revised manuscript (possibly leaving it as an open question - indeed, this is now also explicitly recognized as an interesting avenue for future research in the revised manuscript). Even in the case we might not be able to derive a more general theorem for any $d$ for this work, we find this an important first result in this direction, that we hope will inspire further work.
> > >
> > > We had already indicated this discussion in response to your previous last question about the generalization to higher dimensions. In the revised manuscript (Appendix A.4, lines 841-854) we describe an example of a concave surface in $\mathbb{R}^3$ that resembles a ball with a dent on the north pole, i.e., a crater. This concavity cannot be detected with 0-dimensional PH with respect to any height filtration direction. For the example surface, an interesting direction would be looking at the surface "from the top", but PH would start by seeing a circle, and then the crater itself - always a single connected component. However, 1-dimensional PH with respect to this direction would capture a loop, implying that the surface is concave.
> > >
> > > Alternatively, as we write there, an idea could be to study invariants for multi-parameter persistence in degree $0$, by scanning shapes from multiple directions **simultaneously**. For the example shape, 0-dimensional PH would capture the two connected components with the bi-filtration that looks at the shape from the top, and also from an orthogonal direction from the center of the shape. While the theory and computations for multi-parameter persistence (MPH) are hard, there have been some recent advances, such as the software RIVET [2], or some theoretical insights into $0$-dimensional MPH which would be relevant for this line of work [3].
> > >
> > > [2] The RIVET Developers, RIVET, https://github.com/rivetTDA/rivet/, version 1.1.0, 2020
> > >
> > > [3] On the complexity of zero-dimensional multiparameter persistence, Jacek Brodzki, Matthew Burfitt, Mariam Pirashvili, 2020, preprint available at https://arxiv.org/abs/2008.11532
> > >
> > > Finally, and in response to your earlier comment about the paper only involving data in two dimensions, we must note that the detection of both the number of holes and curvature involved shapes in ($\mathbb{R}^2$ and) $\mathbb{R}^3,$ and with the generalized version of Theorem 1, we now know that this is also possible for the detection of convexity.

---

> ### Author Response · Authors · 2022-08-02
> **Response to Questions 1-3**
>
> > What are the variances of the accuracies in the figures?
>
> This is a valid question. We had indicated in the checklist that ``Due to the large number of computations, the error bars are only reported for some of the experiments (Figure 12 and Figure 13).''  To ensure a fair comparison of the different approaches, we used the same train and test data across all the pipelines. However, we agree that adding error bars would improve the report. We are pursuing further computations towards adding error bars in all our figures and will update them accordingly in the next revision.
>
>
> > Why is PH suited for curvature detection, what is the intuition?
>
> As we mentioned above, we now make the theoretical guarantee for curvature detection more explicit, which hopefully will provide a better intuition to readers. Moreover, in the revised manuscript we now also include an additional Figure 15 precisely with this goal. The figure makes it clear that the length of triangle edges, differ for triangles sampled from manifolds with different curvature. Since the length of the edges is reflected by (in particular 0-dimensional) persistence, the persistence is therefore connected to curvature.
>
> Furthermore, note that the unit disk all have one connected component and no holes, so that there are no long intervals (that reflect this topological information) that can distinguish between them. For this dataset, besides the one long 0-dimensional persistence interval, all intervals are short, and our experiment shows that it is not sufficient to consider only the longest of the short intervals for this problem (p6 lines 216-217 in the revised manuscript).
>
>
> > The paper should be more explicit about the parameters.
>
> Kindly note that we had provided the details about parameters and settings in Appendix A.2 (Appendix B.2 of the revised manuscript) in addition to comprehensive computer code to reproduce our experiments. For PH parameters you mention, we had stated the following: ``For the DTM filtration in Section 3 [detection of the number of holes], we choose $m = 0.03$, so that $0.03 \times 1\,000 = 30$ nearest neighbors are used to calculate the filtration function. To build the filtration in Section 5 (convexity detection), we consider a $20 \times 20$ grid for cubical complexes." (p23 lines 878-881 in the revised manuscript).

---

> ### Author Response · Authors · 2022-08-02
> **Response to Questions 4-5**
>
> > Instead of computing an image from the point cloud in the convexity experiment, could one use scalar field analysis instead? [1]
>
> Thank you for pointing to article [1]. This is an important early theoretical work, and indeed, their main Theorem 2 claims that it is possible to approximate the persistence diagram of any function $f$ and shape $S$ from a point cloud $X$ which is a geodesic dense-enough sample of $S$ using an algebraic construction based on Rips complexes. We had considered to better motivate our choice for cubical over simplicial complexes for convexity detection with an additional figure, and this article strengthens the value of the argument. In the revised manuscript, in Appendix E that provides additional experimental details about convexity, we now therefore include additional two paragraphs and Figure 19 that discusses this choice.
>
> There, we cite the reference [1], and explain how convexity can also be detected by PH with respect to the Vietoris-Rips filtration on point clouds. We start by explaining why the standard Vietoris-Rips would not be successful for this task, and then add that filtering by height (a scalar field, yielding a so called weighted Rips filtration) could help us to detect the disconnected regions in concave shapes. However, as Figure 19 illustrates, these multiple connected components can get connected with an edge (even as soon as they are born) if they are close to each other with respect to euclidean distance. This can be resolved by considering the geodesic distance (the length of the shortest path along the manifold, or a graph), which will allow the multiple connected components to persist longer in the filtration. We conclude stating that the important thing to keep in mind is to choose a filtration that will see disconnected components for concave shapes, and that we choose cubical complexes as they are more straightforward and do not involve the calculation of geodesic distances. Indeed, as the authors of [1] note, ``geodesic distances are not known in advance and have to be estimated through some neighborhood graph distance", and computing full pairwise geodesic distances is expensive (we provide a reference for this claim, and we cite a few deep learning efforts to estimate these geodesic distances on point clouds).
>
> [1] Chazal, Guibas, Oudot, Primoz Skraba. Scalar Field Analysis over Point Cloud Data. Discrete Comput Geom (2011) 46:743–775.
>
>
>  > What is the influence of the point cloud dimension?
>
> Thank you for pointing out this important aspect, we now add Appendix A.4 that focuses on computational complexity. There we provide some results about the theoretical complexity of PH, and explain how it can be extended to higher dimensions for the detection of number of holes and curvature. For the detection of convexity, however, we consider PH with respect to **multiple** height filtration directions. The number of directions grows exponentially with dimension, but one could instead consider (quasi-)random directions. Moreover, the generalization of our pipeline for convexity detection to higher dimensions also involves calculation of PH in higher homology degrees (which focuses not only on the connected components, but also loops, voids, etc), or calculation of 0-dimensional PH with respect to multiple height filtration directions **simultaneously** (multi-parameter persistence). This discussion is detailed in Appendix A.4, and we now also explicitly mention this as an interesting direction for future work.
>
> We think that the added theoretical results (including a new Theorem 1 about convexity detection with PH), and new experiments on real-world data significantly strengthen the contribution, and hope they might motivate you to reconsider your assessment and raise your rating of our manuscript. We hope to have addressed all of your questions satisfactorily, but please let us know if there are any further questions or items that you would like to discuss. We remain attentive to your feedback!

---

### Official Review · Reviewer_786m · 2022-07-12

**Rating:** 5
**Confidence:** 4
**Soundness:** 2 fair
**Presentation:** 3 good
**Contribution:** 2 fair

**Summary:**

In this paper, the author(s) identify special types of problems where persistent homology (PH)  performs well or even better than other state-of-the-art methods in data analysis. They found that PH can work better in three fundamental shape analysis tasks: the detection of the number of holes, curvature and convexity. They have carried out  different experiments to demonstrate the advantage of PH than existing models.

**Questions:**

To demonstrate the "effectiveness" of PH, realistic problems should be considered and a through comparison with the state-of-the-art models should be carried out. In fact, there are many excellent papers, in which these comparison is carried out together with a detailed descriptions of the advantage and limitations of PH models.  For instance,

Z. X. Cang and Guo-Wei Wei, TopologyNet: Topology based deep convolutional and multi-task neural networks for biomolecular property predictions, PLOS Computational Biology, 13(7), e1005690 (2017).

Z. X. Cang, Lin Mu and Guo-Wei Wei, Representability of algebraic topology for biomolecules in machine learning based scoring and virtual screening, PLOS Computational Biology,  14(1), e100592 (2018).

Duc Duy Nguyen, Zixuan Cang, Kedi Wu, Menglun Wang, Yin Cao  and Guo-Wei Wei, Mathematical deep learning for pose and binding affinity prediction and ranking in D3R Grand Challenges, Journal of Computer Aided Molecular Design, 33, 71-82   (2019).

Duc Nguyen, Zixuan Cang, and Guo-Wei Wei, A review of mathematical representations of biomolecular data, Physical Chemistry Chemical Physics, 22, 4343-4367 (2020).

Menglun Wang, Z.  X. Cang, and Guo-Wei Wei,   A topology-based network tree for the prediction of protein-protein binding free energy changes following mutation, Nature Machine Intelligence, 2, 116-123 (2020).

Zhenyu Meng and Kelin Xia, "Persistent spectral–based machine learning (PerSpect ML) for protein-ligand binding affinity prediction." Science Advances, 7 (19), eabc5329 (2021)

Zhao, Qi, Ze Ye, Chao Chen, and Yusu Wang. "Persistence enhanced graph neural network." In International Conference on Artificial Intelligence and Statistics, pp. 2896-2906. PMLR, 2020.

Wong, Chi-Chong, and Chi-Man Vong. "Persistent homology based graph convolution network for fine-grained 3D shape segmentation." In Proceedings of the IEEE/CVF International Conference on Computer Vision, pp. 7098-7107. 2021.

**Ethics Review Area:**

["I don’t know"]

**Limitations:**

As stated above, to demonstrate the "effectiveness" of PH, realistic problems should be considered and a through comparison with the state-of-the-art models should be carried out. In general, it should be easy to show that PH models can characterize certain curvature and convexity related information. However, there are gigantic amount of models that can have much better accuracy in the characterization of curvature and convexity. Only when the PH models are fully compared with state-of-the-art models, the authors can argue about the "Effectiveness of Persistent Homology". Otherwise, it is just the potential application!

**Strengths And Weaknesses:**

The paper has discussed about the advantage of PH in several different areas. It is interesting to see a summerization of these advantages of PH.

PH-based machine learning models have already been widely used in different areas. In fact, the advantage, strength and limitations of PH models have been extensively discussed in various excellent application papers and reviewer papers. The paper lacks a detailed discussion of these models and the proof-of-concept type of experimental tests are not convincing at all!

---

> ### Author Response · Authors · 2022-08-02
> **New experiments on real-world data are included**
>
> Thank you for your careful evaluation of our manuscript and valuable feedback. We are glad you found our discussion about the advantage of PH in several different areas interesting.
>
>
> Your comment about adding experiments with realistic problems is an important one. Indeed, we had listed such experiments tying our investigation with real data as an aim for future research (p9 lines 350-353 in the revised manuscript). For this work, we precisely wanted to avoid complex real world data and instead investigate PH at a fundamental level. There are already many applications of PH to real data (as we had pointed out in the Introduction - p1 lines 20-23 in the revised manuscript, and as is also pointed out in your review), but because in many of these cases the data is complex, there are numerous effects at play and one is often left unsure why PH worked, i.e., what type of topological and/or geometric information it captured that facilitated this. It is precisely for this reason that we decided to study three **fundamental** shape analysis tasks where we have **control** over the source of difference between classes of data (signal). A related observation was made by Reviewer 6onG, who noted that ``there are recently many works integrating PH into various downstream deep learning tasks and achieving state-of-the-art results, but that many works lack insight on why PH can boom performance on these tasks, making the research on what can be detected by PH relevant'', which is precisely what we aim to do in this work. We now describe our scope better in the Introduction (p1 lines 24-28).
>
>
> Thank you for pointing us at a list of works which are all very interesting applications of PH. These works provide further context to the discussion we give in the introduction and we are now adding citations to several of them in the Introduction (p1 lines 20-21, but also, p10 line 359 in the revised manuscript). We may illustrate the above discussion with the first article in the list:
>
> [1] is clearly a very relevant work in molecular biology which provides insights for researchers in the field who want to use PH on similar data (also mentioned by the authors, ``the proposed methods can be easily extended to other applications in the structural prediction of biomolecular properties"). In the abstract the authors state that PH reveals the **hidden** structure relationship in biomolecules and in the conclusions that PH characterizes 3D biomolecular structures. This is further detailed in the paper, where it is stated that, intuitively, 0-dimensional PH describes pairwise atomic interactions, and 1- and 2-dimensional PH characterize the hydrophobic network and geometric rings and voids. The authors also clearly demonstrate that the features in all dimensions contribute to the prediction.
>
> However, it remains elusive what type of structure was important in this application and only limited insights are given on whether PH would work for different applications. This is further complicated by several application-specific aspects; e.g., persistence is associated with the bond length, ring or cavity size, flexibility and steric effect; and combination of PH with CNNs. It is already known from the definition of PH that it reveals connected components, holes and voids. What we are coming to understand in the last years is the important geometry information captured by PH, such as curvature (first by Bubenik) and convexity, as we demonstrate in our work (both experimentally, and now in the revised manuscript, also theoretically, please see Theorem 1). For [1], PH will always reflect the topology (connected components, holes, ...) of the protein structure, but for example, as we demonstrate in our work, for the Vietoris-Rips filtration, the geometry in PH can depict information about their size, or for geodesic distances, also curvature.
>
> [1] Z. X. Cang and Guo-Wei Wei, TopologyNet: Topology based deep convolutional and multi-task neural networks for biomolecular property predictions, PLOS Computational Biology, 13(7), e1005690 (2017).
>
>
> That being said, we do agree that considering real-world data is an important aspect towards understanding the effectiveness of PH in practice. As a step in this direction, we include experiments on the FLAVIA image dataset of plant leaves (Appendix G). The results show that PH is effective in classifying the leaves according to a measure of convexity. We are also adding code for these experiments and in particular a Jupyter notebook for this plant morphology application.

---

> > ### Comment · Reviewer_786m · 2022-08-08
> > **The results are not convincing.**
> >
> > The augment that "our goal was not necessarily to show the advantage of PH over existing models, but rather to demonstrate that PH can solve (i.e., is effective for) certain fundamental problems" is weak!
> >
> > It is not new to know that PH can be used for the so-called "three fundamental shape analysis tasks: the detection of the number of holes, curvature and convexity from 2D and 3D point clouds sampled from shapes"! Further, many other existing models can also be used to solve these so-called "three fundamental shape analysis tasks" (the author also agree with that!)
> >
> > In fact, as mentioned in the previous comments, there are many existing papers which have already addressed "effectiveness" of PH models, in particular, PH-based learning models for molecular data analysis and some special image analysis (not for all image data!). The authors should carefully go though the papers.

---

> > > ### Author Response · Authors · 2022-08-09
> > > **Additional comments about effectiveness**
> > >
> > > > The argument that "our goal was not necessarily to show the advantage of PH over existing models, but rather to demonstrate that PH can solve (i.e., is effective for) certain fundamental problems" is weak!
> > >
> > > > It is not new to know that PH can be used for the so-called "three fundamental shape analysis tasks: the detection of the number of holes, curvature and convexity from 2D and 3D point clouds sampled from shapes"! Further, many other existing models can also be used to solve these so-called "three fundamental shape analysis tasks" (the author also agree with that!)
> > >
> > >
> > > Thank you for your comment. However, we do not quite agree with this representation. For the detection of the number of holes, there are theoretical results that show this is possible (Theorem 6 in Appendix A in the revised manuscript), but, as we write in the Introduction, we are not aware of any experiments that demonstrate this beyond individual examples (e.g., PH on a single or a few point clouds). You seem to suggest that you might know otherwise, in which case we kindly ask that you indicate specific works which focus on how well PH can detect Betti numbers. For curvature, this is indeed true, as we write in our manuscript, but this is a rather new and surprising result from Bubenik (2020), which is, as we write, precisely what motivated us to take a closer look at PH for this geometric notion. Finally, as far as we can tell, the detection of convexity with PH is new: we prove this both in theory (Theorem 1 in Appendix A in the revised manuscript), and demonstrate experimentally (Section 5). Again, if you are aware of previous works in this direction, we would appreciate if you could point us to these.
> > >
> > >
> > > The title of our manuscript is inspired by a famous paper from 1960, "The unreasonable effectiveness of mathematics in the natural sciences", where Wigner discusses, with wonder, how mathematical concepts have applicability far beyond the context in which they were originally developed. In particular, he argues that there is a need to understand **why** such concepts are useful. The same, we believe, is true for persistent homology. While this method has been applied successfully to a wide range of application problems, we believe that for PH to remain relevant, there is a need to better understand why it is so successful.  Thus, we distinguish between the **usefulness** of PH for applications, which has been  attested in hundreds of applications and publications, and its **effectiveness**, namely that PH is capable of producing an intended or desired result. To investigate the effectiveness of PH for applications, we   set out to first identify three fundamental data-analysis tasks.
> > >
> > > PH has been introduced to detect the $k$-dimensional cycles in data, but since there were no previous extensive experimental results on this problem, we decided to start with this most obvious choice for our first task. However, recently, PH has been shown to be applicable beyond this context, when Bubenik showed that PH can detect curvature. Moreover, as we now show, PH can also be used to detect convexity. We find it interesting and remarkable that PH can in fact solve such tasks it is not specifically or uniquely designed for.
> > >
> > > Having established that PH is effective for detecting holes, curvature and convexity, in the revised manuscript we now show that PH is effective in detecting convexity in a real-world data set of leaves. Again, here, our intent was not to show that PH can classify the leaves, since this would have left us with the question as to **why** PH can solve this task. We thus carefully selected a measure of convexity that has been shown to be a good classifier for leaves, and we have shown that we can detect the different values of this measure of convexity for leaves using PH.
> > >
> > > Finally, we do not claim that, for example, PH can better encode curvature than state-of-the-art models that are precisely developed for curvature detection. The benefit of PH over such models would then be that it can reflect **both** topology (i.e., connected components, holes, etc, with longer persistence intervals) **and** curvature (with shorter intervals), which can be useful for many applications (especially if there is no good understanding about the signal in the application). In this way, one would not have to employ specifically developed techniques for each of these particular topological and geometric notions, but could use PH to reveal all this information at the same time. Thank you for raising this question again, we will make sure to include parts of this discussion in our revised manuscript.

---

> > > ### Author Response · Authors · 2022-08-09
> > > **Additional comments about applications**
> > >
> > > > In fact, as mentioned in the previous comments, there are many existing papers which have already addressed "effectiveness" of PH models, in particular, PH-based learning models for molecular data analysis and some special image analysis (not for all image data!). The authors should carefully go through the papers.
> > >
> > >
> > > Indeed, there are many successful and relevant applications of PH: We provide a number of such references in the Introduction, where we had also pointed to a database that lists 395 applications! However, the primary goal of our paper was **not** to add to this important body of literature, but rather, to "take a step back" and investigate the effectiveness of PH at a more fundamental level. The data stemming from applications is very complex, what makes it difficult to understand **why** PH works for that application. In other words, these papers show that PH is useful for a particular application, but they do not focus on understanding why it was, namely the type of topological or geometric information that was captured with PH.
> > >
> > > We did take a close look at all of the references indicated in the initial review, which are also very interesting applications of PH. In our previous response, we even used an example application of PH given in [1] to illustrate our discussion. The complex structure of this data, and the application-specific details make it difficult to identify the reasons behind the success of PH for this problem, and its usefulness on this task does not help us understand if PH would work for other applications. Here, like always, PH detects $k$-dimensional cycles (connected components, holes, voids, ...), but it remains unclear whether there are other geometric notions that were important for this problem, that were also captured with PH and contributed to its good performance.
> > >
> > > For these reasons, it was crucial for us to look at datasets where we could control the signal, i.e., construct the data which precisely differs with respect to the topological or geometric notion of interest (Betti number, curvature, convexity). The new Appendix G then demonstrates that PH has the potential of encoding this information also in real-world applications.
> > >
> > > To conclude, our contribution lies in better understanding **why** PH is so successful in applications, and we believe that this is an important problem to tackle if we want for PH to remain relevant in the future. We hope the above comments provide further clarity about the context and relevance of our work. We thank you again for your valuable time.

---

> ### Author Response · Authors · 2022-08-02
> **Remark on the comparison with the state-of-the-art**
>
> In regard to your comment about the lack of a more detailed comparison with the state-of-the-art (as you note, a gigantic amount of models), we agree that this is a limitation, which had been explicitly acknowledged as such on p9 313-315. At the same time, as we note there, our goal was not necessarily to show the **advantage** of PH over existing models, but rather to demonstrate that PH **can** solve (i.e., is effective for) certain fundamental problems.
>
> We should point out that we do not necessarily expect that on well-specified mathematical tasks PH will beat state-of-the-art algorithms that have been specifically designed for those tasks, particularly detecting convexity, since there is ultimately no free lunch. Instead, what we think is interesting and remarkable is that PH can in fact solve such tasks it is not specifically or uniquely designed for.
> To make this point, we compared different implementations of PH against three baselines, SVMs, MLPs, PointNet, with a multitude of hyperparameters for the baselines and diverse conditions on the data. The reported absolute performance and comparisons make it sufficiently clear that PH can solve these tasks, which is what we claimed. Besides, the new Appendix A in the revised manuscript includes theoretical guarantees that PH can solve these problems, including a new result for the convexity detection. Although we agree that a detailed comparison with state-of-the-art methods is an interesting direction to investigate, our contribution is to identify basic tasks that PH can solve and which can better inform us about why and when PH can be effective in practical tasks.
>
> We hope these responses will better convey the value of our contributions and the approach we have chosen. Also, we believe that the addition of experiments with real data have strengthened our contribution. We hope this might motivate you to raise your assessment of our manuscript. Thanks again for your valuable comments. We remain attentive to your feedback.

---

### Author Response · Authors · 2022-08-02
**Summary of major improvements**

We would like to thank the reviewers for their insightful and constructive comments that helped us reflect critically on our work and make important improvements. There were two major limitations that were raised: the lack of theoretical results (by Reviewers pzbB and 6onG), and the lack of realistic problems (all four reviewers). We submit a revised manuscript that addresses both of these issues, as it includes theoretical results (Appendix A), and an application of PH on a real-world dataset (Appendix G). More precisely, in the revised manuscript, we incorporate these major changes (clearly highlighted in the text) in the following way.


* Firstly, in each of the main Sections 3, 4, 5, we point to a theoretical result that PH can detect respectively the number of holes, curvature, convexity, which are all detailed in the new Appendix A. Theorem 1, about convexity, is a new result. Theorem 6 and Theorem 7, for the detection of the number of holes and curvature, are pointing to the results previously established in the literature. The main contributions in the Introduction and Conclusions now mention Theorem 1. Appendix A also provides a discussion about the theoretical complexity of PH (Appendix A.4), and in particular how it scales with respect to the dimension, as this question was raised by Reviewer pzbB.

* Secondly, Appendix G is added that describes the application of PH to convexity detection on a real-world dataset of images of leaves. These experiments are explicitly mentioned in the main text, in the last sentence of the convexity Section 5.

* Furthermore, some additional improvements have been made throughout the paper to address some of the other questions raised by reviewers. For example, we include two additional figures - Figure 15 and Figure 19, and corresponding accompanying paragraphs to provide some intuition behind curvature detection, and alternative convexity detection with Rips simplicial complex on point clouds (instead of cubical complex on images). Finally, we make our goal and contribution clearer in the Introduction by including some additional references mentioned by the reviewers, and explaining how they differ from (the goals of) our work.

Having in mind that we had received three Borderline accept, with these major improvements, we hope that the reviewers will consider increasing their rating, and that the paper will be considered for publication.

---

### Public Comment · Authors · 2023-01-15
**A comment about the camera ready version**

We would like to start by once again thanking all the reviewers and area chairs for their insightful comments that helped us to significantly improve our work. The key changes have been summarized in our comment ["Summary of major improvements"](https://openreview.net/forum?id=DRjUkfExCix&noteId=a51ilEEkTBK) below, the most important ones being the **addition of theoretical results (Appendix A) and an application of PH on a real-world dataset (Appendix G)**.

However, in the camera ready version, we were able to address some additional comments raised in the discussion:

- As already mentioned, the paper now includes theoretical results that guarantee that PH can detect respectively the number of holes, curvature, convexity (Appendix A). **Theorem 1, about convexity, is a new result, and we have in the meantime extended it from $\mathbb{R}^2$ to $\mathbb{R}^d$. For this, an introduction of a new tubular filtration was essential (for details, see Appendix A.1, and in particular, Definition 1 and Figure 9).** Appendix A.4 also discusses how all PH pipelines can be generalized to higher dimensions. Moreover, these theorems now provide a guarantee that the experimental results generalize to other data.

- As promised in our earlier responses to Reviewer pzbB and Reviewer sPjT, we now report experimental results across multiple runs (Table 2, Table 3, Table 4), and employ batch normalization and regularization for NNs (Appendix B.2).

- We include the two additional references suggested by Reviewer 6onG post rebuttal.


Finally, we address the main remarks from Area Chair hvfd below.

> The title and the main claim need to be adjusted to be more specific/concrete to reflect the actual contribution.

We kept the same title, but have provided more details about our motivation and contribution throughout the Abstract and Introduction, in particular with respect to what we understand with *effectiveness* and how this relates to *usefuless* of PH, and with respect to the lack of a comparison with the state-of-the-art methods for each of the problems. These additional paragraphs summarize our responses to the reviewers who raised these concerns.


> In the main paper, there should be a list of all baseline methods, even though their details can be provided in the supplemental.

The pipelines are summarized in the third paragraph of the Introduction, as well as in the third contribution at the end of this section (and then detailed in Appendix B.2).


> Please comment on whether the baseline methods (e.g., PointNet) have been trained with effective data augmentation (rotation, scaling, etc).

In Appendix B.2, where we describe the methods in detail, including PointNet, we now comment that we do not augment the data during training by randomly rotating the object or jittering position of each point by a Gaussian noise, in order to ensure a fair comparison with the other pipelines.

---

### Meta-Review · Area_Chair_hvfd · 2022-08-27

**Recommendation:** Accept
**Confidence:** Certain

**Metareview:**

This paper focuses on evaluating the learning power of topological features (i.e., persistent homology (PH)) on specific tasks: predicting topological and geometrical properties. On synthetic point cloud data, topological feature outperforms various baseline learning methods when predicting: topology (Betti numbers), curvature and convexity.

This paper received both positive and negative reviews. Both sides have valid points. The major concern is that the experiments are mostly restricted to synthetic settings rather than exploiting the real world problems where PH has shown its learning power -- graph, point cloud data and molecular data. Meanwhile, reviewers do consider the paper valuable as it provides more insights into what can PH effectively predicts and how long/short persistence barcodes should be valued as signal.

After carefully reading the paper, reviews and rebuttals/discussions, We recommend the paper to be accepted. The paper does make a valid contribution by focusing on synthetic data and basic topological and geometric properties. This provides clear message of what PH can predict. It can help the community by providing good insights/guidelines when one tries to incorporate PH into deep learning models.

We do agree with reviewers that the title and the main claim need to be adjusted to be more specific/concrete to reflect the actual contribution. Also the authors should incorporate reviewers' suggestions and strengthen the experiments and references.

Some further suggestions:
In the main paper, there should be a list of all baseline methods, even though their details can be provided in the supplemental. Also please comment on whether the baseline methods (e.g., PointNet) have been trained with effective data augmentation (rotation, scaling, etc).

**Award:**

No

---

### Decision · Program_Chairs · 2022-09-14

Accept